# Structure–Property Relationship of Macrocycles in Organic Photoelectric Devices: A Comprehensive Review

**DOI:** 10.3390/nano13111750

**Published:** 2023-05-27

**Authors:** Chunxiao Zhong, Yong Yan, Qian Peng, Zheng Zhang, Tao Wang, Xin Chen, Jiacheng Wang, Ying Wei, Tonglin Yang, Linghai Xie

**Affiliations:** Center for Molecular Systems and Organic Devices (CMSOD), State Key Laboratory of Organic Electronics and Information Displays, Institute of Advanced Materials (IAM), Nanjing University of Posts & Telecommunications, 9 Wenyuan Road, Nanjing 210023, China; albertzcx@163.com (C.Z.); 13255287602@163.com (Y.Y.); pgqian8963@163.com (Q.P.); siholu@icloud.com (Z.Z.); amazingtao2@163.com (T.W.); xinchen3565@163.com (X.C.); wizardjc@126.com (J.W.)

**Keywords:** macrocycle, organic photoelectric devices, OFETs, OLEDs, OPVs, DSSCs

## Abstract

Macrocycles have attracted significant attention from academia due to their various applications in organic field-effect transistors, organic light-emitting diodes, organic photovoltaics, and dye-sensitized solar cells. Despite the existence of reports on the application of macrocycles in organic optoelectronic devices, these reports are mainly limited to analyzing the structure–property relationship of a particular type of macrocyclic structure, and a systematic discussion on the structure–property is still lacking. Herein, we conducted a comprehensive analysis of a series of macrocycle structures to identify the key factors that affect the structure–property relationship between macrocycles and their optoelectronic device properties, including energy level structure, structural stability, film-forming property, skeleton rigidity, inherent pore structure, spatial hindrance, exclusion of perturbing end-effects, macrocycle size-dependent effects, and fullerene-like charge transport characteristics. These macrocycles exhibit thin-film and single-crystal hole mobility up to 10 and 26.8 cm^2^ V^−1^ s^−1^, respectively, as well as a unique macrocyclization-induced emission enhancement property. A clear understanding of the structure–property relationship between macrocycles and optoelectronic device performance, as well as the creation of novel macrocycle structures such as organic nanogridarenes, may pave the way for high-performance organic optoelectronic devices.

## 1. Introduction

Macrocycles [1], compounds containing a ring of 12 or more atoms, have attracted tremendous attention from theoreticians [2], organic chemists [3,4,5,6], physicists [7], and materials engineers [8] due to their various properties [9] and applications [10,11], such as in organic photoelectric devices. Since the first observation of the macrocyclic compound cyclodextrin by Villiers in 1891, a series of macrocycles, such as cucurbiturils, calixarenes, phthalocyanines, porphyrins, crown ethers, and pillararenes, have been synthesized and extensively reviewed for their applications in host–guest chemistry. However, it was not until 1986, when Cu-phthalocyanine (as electron donor material) was first applied in organic photovoltaics (OPVs) [12], that macrocycles began to be used in the field of organic optoelectronic devices research. In comparison to the similar acyclic molecules or small molecules and polymers, macrocyclic organic semiconductor materials exhibit the following characteristics: (i) Quantum chemical calculations indicate that cyclic compounds possess smaller gaps between the highest occupied molecular orbital (HOMO) and the lowest unoccupied molecular orbital (LUMO) and more extended conjugation than their acyclic analogs. (ii) They exclude perturbing end-effects. (iii) The cyclic nature of these compounds alters their coordination behavior and structural features, which can improve molecular ordering in the solid state and facilitate charge carrier transport. (iv) Macrocyclization significantly reduces the molecular reorganization energy, thereby enhancing field-effect mobilities. (v) The robust backbone and intramolecular interactions of macrocycles inhibit nonradiative relaxation, leading to improved luminous efficiency in organic light-emitting diode (OLED) devices. (vi) Macrocycles can enhance the power conversion efficiency (PCE) of organic photovoltaics (OPVs) by promoting the generation of photogenerated carriers and enhancing carrier transport. As a result, cyclic compounds can be good materials in photoelectric devices, such as organic field-effect transistors (OFETs), OLEDs, OPVs, and dye-sensitized solar cells (DSSCs) (Figure 1). 

Despite the extensive application of macrocycles in the field of organic optoelectronic devices, their performance tends to fall behind that of traditional small-molecule or polymer organic semiconducting materials. To enhance the performance of macrocyclic organic optoelectronic devices, researchers have primarily concentrated on optimizing the following aspects: (i) designing the molecular structure of the active-layer semiconductor macrocycles (planar or nonplanar, conjugated or nonconjugated, etc.), (ii) optimizing the functional layers of the device, (iii) refining device fabrication techniques (vacuum deposition, solution processing, etc.), and (iv) adjusting the device configuration. Undoubtedly, the design of macrocyclic molecular structures aimed at improving device performance is critical. Therefore, in this article, our primary focus is on exploring the structure–property relationships between macrocycles and their optoelectronic device performance. Specifically, we comprehensively discuss these relationships from multiple perspectives, including energy level structure, structural stability, film-forming property, skeleton rigidity, inherent pore structure, spatial hindrance, exclusion of perturbing end-effects, macrocycle-size-dependent effects, and fullerene-like charge transport characteristics.

Previously, Kesters et al. [13] and Gao et al. [14] extensively reported on the research progress of porphyrins in OPVs in 2015 and 2020, respectively. Furthermore, starting from 2010, several comprehensive reviews have been published on the application of porphyrins, phthalocyanines, and calix[4]arene-based macrocyclic dyes in DSSCs [15,16,17,18]. In this review, we present the research progress on organic macrocycles in optoelectronic devices and discuss in detail the structure–property relationship between macrocycle structures and their device performance. In Section 2 and Section 3, we provide a comprehensive summary of the research progress of macrocyclic molecules in the field of OFETs and OLEDs since the 1990s. In Section 4 and Section 5, we focus on summarizing the research on representative macrocyclic structures in the field of OPVs and DSSCs since 2009.

## 2. Application of Macrocycles in OFET Device

Organic field-effect transistors (OFETs) [19,20,21] are devices composed of an organic semiconducting layer, a gate insulator layer, and three terminals (drain, source, and gate electrodes). OFETs are not only important building blocks for low-cost and flexible organic circuits but also provide valuable insight into the charge transport of π-conjugated systems. Therefore, OFETs serve as effective tools for exploring the relationship between the structure and properties of macrocycles, including the parameters of field-effect mobility (*μ*), current on/off ratio (*I*_on_/*I*_off_), and threshold voltage (*V*_T_). OFETs can be categorized into four different types, as illustrated in Figure 2: (a) the bottom gate top contact, (b) the bottom gate bottom contact, (c) the top gate top contact, and (d) the top gate bottom contact. Notably, the performance of devices (a) and (d) is generally higher than that of devices (b) and (c) due to the improved contact between the organic semiconductor layer and the electrodes. Organic semiconductor materials can be classified as either p-type or n-type, depending on which type of charge carrier is more efficiently transported through the material. When both holes and electrons can be injected and transported in the same device, the device exhibits ambipolar characteristics. Undoubtedly, the device performance is influenced by various factors, such as device fabrication techniques, the device configuration, the semiconductor materials, the electrodes, and the gate insulator materials [20]. In this section, we mainly focus on the structure–property relationship between the macrocycle structure and the optoelectronic device performance. In OFET devices, macrocyclic organic semiconductor molecules with a large π-conjugated system and significant intermolecular π–π overlap are key to achieving high carrier mobility. Typically, device performance can be enhanced by introducing substituents at the peri positions of the macrocycle, introducing polarity, increasing the C/H ratio, or adding heteroatoms to generate hydrogen bonds, halogen–halogen interactions, or chalcogen–chalcogen interactions. Moreover, studying single-crystal field-effect transistors is beneficial for achieving high device performance and gaining a deeper understanding of the conduction mechanism of organic macrocyclic semiconductors. Finally, the charge carrier mobilities of representative macrocycles in OFETs are shown in Figure 3.

### 2.1. p-Type Macrocycle Semiconductors

#### 2.1.1. Phthalocyanine

Phthalocyanine (Pc) (**1**) is an aromatic macrocyclic system with 18 π electrons, which is composed of four benzopyrroles linked together with nitrogen atoms. The central cavity has a diameter of about 0.27 nm and can accommodate many metallic and nonmetallic elements. Phthalocyanines and derivatives (Figure 4) are widely used in organic field-effect transistors (OFETs) due to their structural diversity, ease of synthesis, excellent thermal and chemical stability, as well as nontoxicity. Usually, the metal phthalocyanine (MPc) formed by phthalocyanine and metal cations still maintains excellent planarity (e.g., CuPc(**2a**), ZnPc(**2b**), FePc(**2c**), CoPc(**2d**), NiPc(**2e**), and MnPc(**2f**)), which is conducive to the formation of tight π–π packing to improve carrier transport. Among them, **2a** is probably the first material in the phthalocyanine structure to be used in the study of OFETs. Depending on the dielectric material, substrate temperature [22,23,24,25,26], and channel length to width (L/W) ratios [25], the reported transistors mobility of **2a** thin film ranges from 10^−5^ to 1.3 cm^2^ V^−1^ s^−1^, and the current on/off ratio ranges from 10^2^ to 10^5^. However, the deposited film may decrease transistor mobility and device stability in air due to the occurrence of high-density grain boundaries. The highly ordered copper phthalocyanine films with large crystalline domains obtained by weak epitaxy growth (WEG) can significantly improve device mobility and stability in air [27,28]. Compared with the thin-film devices, which have a large fluctuation in device performance, the mobility of **2a** single-crystal transistors [29,30,31,32] is basically maintained between 0.1 and 1 cm^2^ V^−1^ s^−1^, which is comparable to or even better than that of amorphous silicon, and the current on/off ratio is maintained between 10^4^ and 10^6^. However, the **2a** single-crystal transistors can achieve lower threshold voltages (−0.2 V) [31] than thin films (−2 V) [26]. The mobilities of OFETs for phthalocyanines and asymmetrically substituted phthalocyanines are only 10^−3^ cm V^−1^ s^−1^ [23] and 10^−6^ cm^2^ V^−1^ s^−1^, respectively [33]. When the source/drain electrodes of an OFET sandwiched between the first active layer of **2a** and the second active layer of cobalt phthalocyanine **2d**, the mobility increases from 0.04 to 0.11 cm^2^ V^−1^ s^−1^ in comparison with that of a **2a** top-contact device (Table 1) [34]. However, for traditional metal phthalocyanines, large-area, low-cost solution-processing methods cannot be used to fabricate OFET devices due to their poor solubility. The sodium salts of sulfonated **2d** (CoPcSx) not only have good solubility; also, the mobility of OFET devices fabricated via the spin-coating method is as high as 0.2 cm^2^ V^−1^ s^−1^, which is much higher than that of devices fabricated via traditional vacuum evaporation (10^−4^ cm^2^ V^−1^ s^−1^) [35]. Moreover, the introduction of substituents at the *α* and *β* positions of phthalocyanine is an effective method to improve solubility. Ilgın Nar et al. reported an OFET device based on *α*-*n*-butoxy, *β*-ethynylphenyl-substituted cobalt (II) Pc (**3**) spin-coated films as the active layer and obtained a hole mobility of 1.12 × 10^−2^ cm^2^ V^−1^ s^−1^ [36]. Similarly, OFET devices based on substituent-modified copper phthalocyanine (**4a**), nickel phthalocyanine (**4b**), and cobalt phthalocyanine (**4c**) spin-coated thin films exhibited the highest mobilities of 9.77 × 10^−2^ cm^2^ V^−1^ s^−1^, 6.12 × 10^−4^ cm^2^ V^−1^ s^−1^, and 2.32 × 10^−2^ cm^2^ V^−1^ s^−1^ [37], respectively. By using the WEG method, a film device made of **2b** deposited on a para-sexiphenyl (*p*-6P) self-assembly layer showed a hole mobility as high as 0.32 cm^2^ V^−1^ s^−1^ [38], which is at the same level as that of its single crystal transistor (0.75 cm^2^ V^−1^ s^−1^) [39]. Some **2b** spin-coated thin-film transistor devices based on ferrocenylcarborane functionalization exhibited a hole mobility of 0.47 cm^2^ V^−1^ s^−1^, which is higher than that of a similar unfunctionalized one (0.027 cm^2^ V^−1^ s^−1^) [40].

Different from the planar MPcs, such as **2a**, titanyl phthalocyanine (TiOPc) (**5a**), vanadium phthalocyanine (VOPc) (**5b**), and lead phthalocyanine (**2i**) are V-shaped and polar molecules. Although three crystal structures of TiOPc, monoclinic phase I (*β*-**5a**), triclinic phase II (*α*-**5a**) and triclinic phase Y, were reported, *α*-**5a** showed significant molecular overlap and ultra-close π stacking. The mobility of *α*-**5a** thin-film transistor devices based on octadecyltrichlorosilane (OTS)-modified Si/SiO_2_ substrates is as high as 3.31 cm^2^ V^−1^ s^−1^, and the highest mobility reached 10 cm^2^ V^−1^ s^−1^ [41]. Furthermore, the single-crystal transistors of *α*-**5a** exhibited an average mobility of 10.6 cm^2^ V^−1^ s^−1^ and a maximum mobility of up to 26.8 cm^2^ V^−1^ s^−1^ [42]. For **5b**, the mobility of thin-film OEFT devices based on OTS-modified Si/SiO_2_ substrates (0.3–1 cm^2^ V^−1^ s^−1^) is much higher than that of devices based on bare Si/SiO_2_ substrates (0.003–0.006 cm^2^ V^−1^ s^−1^) [43]. Moreover, the mobility of the devices based on *p*-6P [44] and 5,5‴-bis(4-fluorophenyl)-2,2′:5′,2″:5″,2‴-quaterthiophene (F2-P4T) [45] were further improved to 1.23 cm^2^ V^−1^ s^−1^ and 2.6 cm^2^ V^−1^ s^−1^, respectively. Similarly, OFET devices utilizing **2i** film epitaxial growth on a monolayer of 5,5″-bis(3′-fluoro-biphenyx-4-YL)-2,2′:5′,2″-terthiophene (M-F2BP3T) at 175 °C as the active layer exhibit hole mobility (0.05–0.31 cm^2^ V^−1^ s^−1^) that are 1–2 orders of magnitude higher than those observed for devices based on bare Si/SiO_2_ substrates [46]. By using a copper hexadecafluorophthalocyanine (F16CuPc)(**6**)/**2a** heterojunction unit as a buffer layer and F2-P4T-modified Si/SiO_2_ substrates, OFET devices with weak-epitaxy-growth **3**/**5b** films exhibited hole mobility at 3.65 cm^2^ V^−1^ s^−1^ (with the highest mobility at 4.08 cm^2^ V^−1^ s^−1^), an on/off current ratio at 3.76 × 10^6^, and threshold voltages at −2.66 V [47]. However, thin-film transistors based on tetraoctyl-substituted **7** spin-coating film showed a hole mobility of 0.017 cm^2^ V^−1^ s^−1^ [48]. It is worth mentioning that Yan et al. reported flexible **7** thin-film transistors produced by using the WEG method. The devices not only showed a mobility of 0.5 cm^2^ V^−1^ s^−1^ but also exhibited recoverable electrical behavior when the strain was less than 1.5% [49].

Compared with monomeric MPc, the bis(phthalocyaninato) rare-earth derivatives exhibit better film-forming behavior owing to their better solubility. The transistor mobilities of Lu(Pc)[Pc(OC_8_H_17_)_8_] (**8a**) (1.7 × 10^−3^ cm^2^ V^−1^ s^−1^) produced based on the Langmuir–Blodgett (LB) technique is about three times higher than that of Tb(Pc)[Pc(OC_8_H_17_)_8_] (**8b**) (about 6.4 × 10^−4^ cm^2^ V^−1^ s^−1^), probably due to the closer stacking of the **8a** molecules in the film [50]. Double-decker complexes Tb(III) and Dy(III) phthalocyanine (**9a** and **9b**) film transistors were fabricated by evaporation on hexamethyldisilazane (HMDS, 500 nm)-treated Si/SiO_2_ substrates [51]. Interestingly, **9a** showed a hole mobility of about 10^−4^ cm^2^ V^−1^ s^−1^, while **9b** exhibited an air-stable ambipolar behavior with a hole mobility of about 10^−4^ cm^2^ V^−1^ s^−1^ and an electron mobility of about 10^−5^ cm^2^ V^−1^ s^−1^. By using poly(ethylene terephthalate) (PET) as a flexible substrate, a highly oriented and solution-processed film transistor of bis(phthalocyaninato) europium complex [Pc(OAr)_8_]Eu[Pc(OAr)_8_] (**10**) [52] showed ambipolar characteristics with a hole mobility of 0.10 cm^2^ V^−1^ s^−1^ in air and 0.01 cm^2^ V^−1^ s^−1^ for electrons under nitrogen conditions. In addition, the OFET devices of three amphiphilic tris(phthalocyaninato) rare-earth triple-decker complexes (**11a**, **11b**, and **11c**) as the active layer exhibited hole mobilities as high as 0.60, 0.40, and 0.24 cm^2^ V^−1^ s^−1^, respectively, due to the intramolecular π–π stacking and the *J* aggregation in the LB films [53]. Similarly, an amphiphilic (phthalocyaninato) (porphyrinato) Eu(III) triple-decker complex [Pc-(OPh)_8_]Eu[Pc(OPh)_8_]Eu[TP(C≡CCOOH)PP] (**12**) was synthesized and characterized under ambient conditions [54]. The OFET devices fabricated via the phase-transfer method with one-dimensional nanoribbons morphology displayed hole and electron mobilities of 0.11 and 4 × 10^−4^ cm^2^ V^−1^ s^−1^, respectively, which were 3–6 orders higher than those of devices with nanoparticle morphology fabricated via the quasi-Langmuir–Shäfer (QLS) technique. It is worth mentioning that the single-crystal OFET devices made of dimeric phthalocyanine involving triple-decker complex ([Pc(SC_6_H_13_)_8_]_2_Eu_2_[BiPc(SC_6_H_13_)_12_]Eu_2_[Pc(SC_6_H_13_)_8_]_2_) (**13**) [55] showed hole and electron mobilities of up to 18 and 0.3 cm^2^ V^−1^ s^−1^, respectively. In addition to double-/triple-decker MPc, ball-type dinuclear M_2_Pc_2_ also exhibits good solution processing properties. By using the spin-coated films of Co_2_Pc_2_ (**14a**), Zn_2_Pc_2_ (**14b**), and Cu_2_Pc_2_ (**14c**) as the active semiconductor layer, these transistors’ mobilities range from 5.2 × 10^−3^ to 6.8 × 10^−2^ cm^2^ V^−1^ s^−1^ [56,57].

**Table 1 nanomaterials-13-01750-t001:** Representative performance data of phthalocyanine in p-type OFETs.

Material	LUMO (eV)	HOMO (eV)	*µ* (cm^2^ V^−1^ s^−1^)	*I*_on_/*I*_off_	*V*_T_/V	Ref.
**2a**			1.3 ± 0.02	3.08 × 10^5^	2	[25]
**2b**			0.75	7.34 × 10^3^		[39]
**2b/p-6P**			0.32			[38]
**2d**			0.2	10^3^	−0.8	[35]
**3**			1.12 × 10^−2^			[36]
**4a**			9.77 × 10^−2^	10^3^	11	[37]
**4b**			6.12 × 10^−4^	10^2^	57	[37]
**4c**			2.32 × 10^−2^	10^4^	35	[37]
**5a**			26.8	10^4^–10^7^		[42]
**5b**			0.3–1.0	10^6^–10^8^		[43]
**5b/p-6p**			1.23	10^6^		[44]
**5b/F2-P4T**			1.2–2.6	10^6^–10^7^	−1 to −5	[45]
**6/2a**			4.08	3.76 × 10^6^	−2.66	[47]
**8a**	−5.38	−6.45	1.7 × 10^−3^			[50]
**8b**	−5.27	−6.43	6.4 × 10^−4^			[50]
**9a**			10^−4^			[51]
**9b**			10^−4^			[51]
**10**	−3.41	−4.63	0.1			[52]
**11a**			0.6			[53]
**11b**			0.4			[53]
**11c**			0.24			[53]
**12**			0.11		−8	[54]
**13**			18	10^3^–10^4^		[55]
**14a**			5.2 × 10^−3^		−2.5	[56]
**14b**			6.7 × 10^−3^		−4.0	[56]
**14c**			4.4 × 10^−2^		−27.6	[56]

#### 2.1.2. Porphyrin

Porphyrins, a class of macrocyclic structures that is similar to phthalocyanines, have also been widely applied in OFET devices. The crystalline film of 2,3,7,8,12,13,17,18-octaethylporphyrinato platinum (PtOEP) (**15a**) (Figure 5) epitaxially grown via thermal evaporation was fabricated into OFET devices with a mobility of 1.8 × 10^−3^ cm^2^ V^−1^ s^−1^ (Table 2) and an on/off current ratio of 10^4^ [58]. After that, four kinds of 2,3,7,8,12,13,17,18-octaethyl-porphyrinato metal complexes (**15b**, **15c**, **15d**, and **15e**) single crystal were used in OFET devices as the active layer, and the hole mobilities increased to 0.2, 0.068, 0.036, and 0.014 cm^2^ V^−1^ s^−1^, respectively [59]. Transistors based on tetrabenzoporphyrin (**16a**) exhibited mobilities of 0.017 cm^2^ V^−1^ s^−1^ [60], and the devices made of polycrystalline **16a** thin-film also showed the same level mobilities at 0.01 cm^2^ V^−1^ s^−1^ [61,62]. However, devices made of organometallic tetrabenzoporphyrin (NiTBP (**16b**) and CuTBP (**16c**)) thin film fabricated via the spin-coating technique demonstrated improved field-effect performance with hole mobilities at 0.2 and 0.1 cm^2^ V^−1^ s^−1^, respectively [63,64]. For 5,10,15,20-tetra-phenyl porphyrin (**17a**), an OFET device with spin-coated films showed mobility as high as 0.012 cm^2^ V^−1^ s^−1^ [65]. Additionally, single-crystal devices based on tetra(phenyl)porphyrin derivatives (**17b**,**17d**,**18b**) demonstrated mobilities of 1.8 × 10^−3^, 6.2 × 10^−2^, and 0.32 cm^2^ V^−1^ s^−1^ [66,67], respectively. Using the π-extended strategy, single-crystal OFETs made of four 2-ethynyl-5-hexylthiophene peripheral-arms-substituted porphyrin derivatives (H_2_TP (**17c**) and ZnTP (**18a**)) provided mobilities as high as 0.85 and 2.90 cm^2^ V^−1^ s^−1^, respectively, which are higher than those of film devices [68]. Similarly, OFETs of the π-extended porphyrin derivatives (2TBPH (**19a**), 2TBPZ (**19b**), TEPP (**20a**), DTEPP (**20b**), **20c**, **20d**, PD-1 (**21**), H2TPEP (**22a**), and ZnTPEP (**22b**)) displayed high hole mobilities ranging from 0.026 to 2.57 cm^2^ V^−1^ s^−1^ [69,70,71,72,73]. For double-thiophene-substituted porphyrin derivatives (H2DTP (**23a**), NiDTP (**23b**), and CuDTP (**23c**)) [74], their single-crystal OFETs exhibited mobilities of up to 0.15, 1.50, and 0.74 cm^2^ V^−1^ s^−1^, respectively. Compared with **23a**, the dramatic increase in the mobilities of **23b** and **23c** is probably due to their denser molecular arrangement. Furthermore, a series of metal-free porphyrin derivatives with functionalized triarylamines at the *meso* position were also reported (**24a**, **24b**, **24c**, **24d**, and **24e**). By using solution-processable technique, these molecules showed excellent hole mobilities [75] of 0.66 (*I*_on/off_ = 10^8^), 0.25 (*I*_on/off_ = 10^7^), 3.74 (*I*_on/off_ = 10^8^), 0.72 (*I*_on/off_ = 10^6^), and 4.40 (*I*_on/off_ = 10^7^) cm^2^ V^−1^ s^−1^, respectively. The high mobilities of **24c** and **24e** could be attributed to the presence of electron-withdrawing fluorine and trifluoromethyl groups. A class of extended porphyrin analogue cyclo[8]pyrrole (**25**) [76] and cyclo[6]pyrrole (**26**) [77] has also been reported. These molecules, with highly extended *π* systems and nearly coplanar conformation, increase the intermolecular overlap of *π*–*π* orbitals in the solid states, leading to high mobility. Their OFETs using LB films as the active layer demonstrated high hole mobilities at 0.68 and 0.014 cm^2^ V^−1^ s^−1^.

#### 2.1.3. Thiophene-Containing Macrocycles

Thiophene-based conjugated materials have been widely used in OFET semiconductor layers due to their chemical stability, thermal stability, ease of modification, and extraordinary electronic properties. However, compared with the corresponding linear oligomers and polymers, the cyclic structures exhibit a well-defined structure, infinite *π*-conjugated chains, and higher symmetry, which may improve the *π*–*π* interactions and therefore facilitate charge carrying. A class of thiophene-containing macrocycles (THCMs), tetrathia[22]annulene[2,1,2,1] (**27a**) (Figure 6) with a planar or near-planar structure, was systematically investigated as the OFET semiconductor layer. Devices made of **27a [78]** based on vapor deposition of a film on OTS-modified Si/SiO_2_ substrate showed an average mobility of 0.02 cm^2^ V^−1^ s^−1^ (Table 3), and the maximal hole mobility was as high as 0.05 cm^2^ V^−1^ s^−1^ (*I*_on/off_ = 1.1 × 10^3^) which is comparable to that of linear oligothienylenevinylenes (0.05 cm^2^ V^−1^ s^−1^) [79]. Then, three kinds of *meso*-substituted tetrathia[22]annulene[2,1,2,1] macrocycles (**27b**, **27c**, **27d**) [80] were fabricated into OFET via the vapor deposition technique, with the highest mobility found for **27c** of 0.63 cm^2^ V^−1^ s^−1^. However, the film transistors of electronegative-groups-substituted **27c** derivatives (**27d**, **27e**, and **27f**) [81,82] provided hole mobilities of 0.23 (*I*_on/off_ = 5 × 10^5^), 0.012 and 0.73 cm^2^ V^−1^ s^−1^ (*I*_on/off_ = 1.4 × 10^7^), respectively. The performance of all these devices based on tetrathia[22]annulene[2,1,2,1] and its derivatives demonstrated that the substituted groups have significant influences on both the charge mobilities and on/off ratios. Moreover, the devices containing furan analogues **28a** and **28b** [83] showed hole mobilities of 0.35 (highest mobility at 0.40 cm^2^ V^−1^ s^−1^) and 0.11 cm^2^ V^−1^ s^−1^, respectively. In addition to tetrathia[22]annulene[2,1,2,1] cyclic structures (**29**), macrocycles containing ethynylene linkages have also been applied in OFETs via the spin-coating method and afforded hole mobilities of 10^−5^–10^−4^ cm^2^ V^−1^ s^−1^ [84,85]. Thin-film transistors based on a diketopyrrolopyrrole (DPP)-conjugated macrocycle showed ambipolar transport with hole mobilities of 3.4 × 10^−4^ cm^2^ V^−1^ s^−1^ and 2.3 × 10^−4^ cm^2^ V^−1^ s^−1^ for electrons [86].

#### 2.1.4. Triarylamine-, Carbazole-, and Acene-Containing Macrocycles

Triarylamine derivatives have been widely applied in the optoelectronic field in OFET applications. However, these linear, star-shaped, or dendrimeric structures usually show low carrier mobility due to their amorphous nature in the film states. In contrast, the confinement of the rotation of the phenyl group via macrocyclization improves the planarity of the molecular skeleton and the tightly ordered packing of the molecules, which is an effective method to improve charge transport. Therefore, a triarylamine-containing macrocycle (TACM) (**30**) (Figure 6) [87] was developed by Zhu et al. The vapor-deposited thin films with a highly crystalline layer-by-layer packing structure of **30** exhibited a high mobility of 0.015 cm^2^ V^−1^ s^−1^ (Table 4) and an on/off ratio of 10^7^. In contrast, the devices made of the linear analogue only showed a hole mobility of 2 × 10^−4^ cm^2^ V^−1^ s^−1^ because the films were almost amorphous. Additionally, single-crystal transistors of **30** [88] were fabricated and examined, showing a mobility of up to 0.05 cm^2^ V^−1^ s^−1^. It is worth mentioning that the mobility of single-crystal devices is heavily dependent on the crystal size used. The size effect can be tentatively explained by the following three factors: (i) smaller crystals are higher-quality; (ii) smaller/thinner crystals have a better physical contact with the substrate; (iii) larger contact resistance probably exists in devices made of big/thick crystals.

Similar to cyclic triarylamine derivatives, carbazole-containing macrocycles (CZCMs) have also been applied in OFET devices. Multicrystalline thin films of **31a** and **31b** [89] were obtained via the vapor-deposition technique on OTS-treated Si/SiO_2_ substrate, and the **31b** devices showed the highest mobility of 0.013 cm^2^ V^−1^ s^−1^. However, the linear analogue only gave amorphous thin films with a low mobility of 3 × 10^−4^ cm^2^ V^−1^ s^−1^. The huge performance difference between the cyclic and linear structures was mainly attributed to the following two factors: (i) the cyclic structure is more conducive to the molecular ordering and *π*–*π* stacking to form polycrystalline films; (ii) macrocyclization significantly reduces the molecular reorganization energy. Moreover, a class of [4]cyclo-*N*-alkyl-2,7-carbazoles nanorings was also used as the active layer in an OFET, as reported by Poriel et al. The nonoptimized transistors based on vapor-deposited thin films of [4]C-Et-carbazole (**32a**) exhibited typical p-type performance with a mobility of 1.1 × 10^−5^ cm^2^ V^−1^ s^−1^ [90]. Although its hole mobility is significantly lower than that of the state-of-the-art transistors, this work first used nanorings in OFET devices. Then, Poriel et al. investigated the structure−properties relationships of nanoring series in detail for the first time [91]. The hole mobilities of **32a**, [4]C-Bu-carbazole (**32b**), and [4]C-Hex-carbazole (**32c**) are very close, at 1.03 × 10^−5^ cm^2^ V^−1^ s^−1^, 1.04 × 10^−5^ cm^2^ V^−1^ s^−1^, and 8.81 × 10^−6^ cm^2^ V^−1^ s^−1^, respectively while the benchmark [8]CPP does not exhibit any field-effect mobility. Another carbazole-containing macrocycle is the cyclic carbazolylacetylene derivative. The single-bundle nanofibers transistors based on organogelator compound **33** [92], produced using a simple method, demonstrated an average mobility of 1.3 × 10^−3^ cm^2^ V^−1^ s^−1^ and a highest mobility of 3.6 × 10^−3^ cm^2^ V^−1^ s^−1^, which is two orders of magnitude larger than that of thin-film devices.

In addition to triarylamine and carbazole macrocyclic derivatives, acene-containing macrocycles (ACMs), such as anthracene and phenanthrene, have also been used in organic semiconductors. Miao et al. reported a transistor fabricated from cyclic anthracene derivative (**34**) [93] by using thermal evaporation, and the highest hole mobility was up to 0.07 cm^2^ V^−1^ s^−1^. For cyclic phenanthrene derivatives (**35**, **36**, and **37**) [94], the solution-processed thin-film transistors displayed filed-effect mobilities of 7.0 × 10^−4^, 2.4 × 10^−4^, and 4.4 × 10^−5^ cm^2^ V^−1^ s^−1^, respectively. However, a film of **35** prepared with phenyltrichlorosilane-modified Si/SiO_2_ substrate contained broken birefringent fibers, and the mobility could be further improved to 1.2 × 10^−3^ cm^2^ V^−1^ s^−1^.

**Table 4 nanomaterials-13-01750-t004:** Representative performance data of triarylamine-, carbazole-, and acene-containing macrocycles in p-type OFETs.

Material	LUMO (eV)	HOMO (eV)	*µ* (cm^2^ V^−1^ s^−1^)	*I*_on_/*I*_off_	V_T_/V	Ref.
**30**			1.5 × 10^−2^	10^7^		[87]
**31a**		−5.09	5.3 × 10^−3^	10^6^	−4.7	[89]
**31b**		−5.22	1.3 × 10^−2^	10^7^	−22.0	[89]
**32a**	−2.22	−5.17	2.71 × 10^−4^		−13.1	[91]
**32b**	−2.40	−5.18	2.78 × 10^−4^		−12.8	[91]
**32c**	−2.21	−5.19	1.37 × 10^−4^		−13.9	[91]
**33**			3.6 × 10^−3^	10^5^		[92]
**34**			0.05	10^4^		[93]
**35**	−2.31	−5.32	7.0 × 10^−4^			[94]
**36**	−2.42	−5.54	2.4 × 10^−4^			[94]
**37**	−1.92	−5.45	4.4 × 10^−5^			[94]

### 2.2. n-Type Macrocycle Semiconductors

Both p-type and n-type semiconductor materials are essential for ambipolar transistors, p–n junctions, and organic complementary circuits. However, in contrast to the large number of p-type semiconductor materials with a hole mobility above 10 cm^2^ V^−1^ s^−1^, there are still only a few ambient-stable n-type semiconductor materials with an electron mobility above 1 cm^2^ V^−1^ s^−1^ that have been reported. Typically, n-type semiconductors have the lowest unoccupied molecular orbital (LUMO) levels to facilitate electron injection from electrodes. However, the environmental stability of a metal electrode with a low work function (Al, Mg, etc.) is much lower than that of their high-work-function counterparts, such as Au and Pt. Although the problem of the high injection barrier between semiconductor materials and high-work-function electrodes can be solved through electrode modifications and electrode–semiconductor interface engineering, the influence of n-type semiconductor stability on device stability cannot be changed. Therefore, reducing the LUMO level via molecular design is a more effective way to obtain environmentally stable n-type OFET devices. The molecular design strategies usually used to reduce the LUMO level mainly include introducing electron-withdrawing groups or constructing a donor–acceptor structure, acceptor–acceptor structure, ladder structure, and so on. In recent years, research results have shown that the combination of the above design strategy and macrocyclization can further effectively reduce the molecular LUMO level [95]. In this section, we introduce n-type organic semiconductors based on phthalocyanine and cyclic perylene diimide (PDI) derivatives.

#### 2.2.1. Phthalocyanine

Although most phthalocyanine derivatives exhibit p-type semiconductor behavior, electron carrier transport can be achieved by introducing electron-withdrawing groups, such as fluoride and chloride. To date, a series of phthalocyanine derivatives with multiple electron-withdrawing groups have been applied in air-stabile n-channel OFET devices. Among them, the most representative is copper hexadecafluorophthalocyanine (F_16_CuPc) (**6**) [96]. The highly ordered, deposited, thin-film transistors demonstrated electron mobility as high as 0.03 cm^2^ V^−1^ s^−1^, while the other metallophthalocyanines analogues, such as zinc hexadecafluorophthalocyanine (F_16_ZnPc) (**38a**) (Figure 7), cobalt hexadecafluorophthalocyanine (F_16_CoPc) (**38b**), and iron hexadecafluorophthalocyanine (F_16_FePc) (**38c**), only showed mobilities ranging from 10^−5^ to 10^−3^ cm^2^ V^−1^ s^−1^. In comparison, single-crystal transistors of **6** [97] based on bare Si/SiO_2_ substrates exhibited a higher mobility of 0.2 cm^2^ V^−1^ s^−1^ with an on/off ratio at about 6 × 10^4^. By using OTS-modified SiO_2_ as the dielectric layer, the mobilities of single crystals **38a**, **38b**, and **6** were further improved to 1.1, 0.8, and 0.6 cm^2^ V^−1^ s^−1^ [98], respectively. Furthermore, for single-crystal OFET devices of **6**, Kloc et al. observed that the mobility increased with decreasing crystal thickness, and the increase was more pronounced when the thickness was less than a few hundred nanometers [99]. Compared with fluorinated molecules, chlorinated molecules exhibit lower LUMO levels, probably because chlorine contains empty 3d orbitals that can accept π electrons [100]. Therefore, n-type air-stable OFET have been fabricated by using copper hexachlorophthalocyanine (Cl_16_CuPc) (**38d**) as the semiconductor layer [100,101], and the highest electron mobility reached 0.11 cm^2^ V^−1^ s^−1^ with an on/off ratio at 1 × 10^5^. Similarly, thin-film transistors of vanadyl hexachlorophthalocyanine (Cl_16_VOPc) (**39a**) [102] showed low mobilities of (2.0 ± 0.1) × 10^−3^ cm^2^ V^−1^ s^−1^ (Table 5) due to the poor ordering in films. An axially oxygen substituted tin (IV) phthalocyanine oxide (SnOPc) (**39b**) [103] was also examined, and the devices based on a *p*-6P-modified substrate showed a mobility of up to 0.44 cm^2^ V^−1^ s^−1^. The film transistors of phthalocyanato tin (IV) dichloride (SnCl_2_Pc) (**40**) based on *p*-6P-modified substrate provided the highest mobility of 0.3 cm^2^ V^−1^ s^−1^ due to the intermolecular π–π stacking in a direction parallel to the substrate [104]. Using the solution process, the mobilities of axially substituted (OR)_2_-SnPcs (**41**) were located at (0.02–0.7) × 10^−2^ cm^2^ V^−1^ s^−1^ [105].

Recently, a series of silicon phthalocyanines (SiPcs) derivatives have been synthesized and applied in OFETs. In 2018, Lessard et al., for the first time, reported three axially substituted SiPc molecules (**42a**, **42b**, and **42c**) [106] as n-type semiconductor materials with electron mobilities of 9.3 × 10^−4^, 3.3 × 10^−4^, and 5.6 × 10^−5^ cm^2^ V^−1^ s^−1^, respectively, in a vacuum. By using a trichloro(octadecyl)silane (ODTS)-modified substrate and heating to 200 °C, the field-effect mobility of molecule **42a** was further increased to 1.35 × 10^−2^ cm^2^ V^−1^ s^−1^. After that, a large number of axially substituted SiPc derivatives [107,108,109] with a low mobility (10^−5^–10^−3^ cm^2^ V^−1^ s^−1^) and high threshold voltage (20–50 V) have been reported in OFETs. By using low-work-function metal manganese as the contact interlayer, F10-SiPc (**42d**) devices showed a low average threshold voltage of 7.8 V [110]. The solution-processed OFET of bis(tri-n-butylsilyl oxide) SiPc [111], with manganese as the contact interlayer, exhibited the highest mobility of 0.028 cm^2^ V^−1^ s^−1^. Moreover, Lessard et al. found that the threshold voltage (*V*_T_) of OFETs can be reduced by introducing electron-withdrawing groups based on phenoxy-substituted SiPcs (**42a**, **42b**, **42i**, **42j**, **42k**, **42l**, **42m**, **42n**, **42o**, and **42p**) [111]. Thus, phenoxy-SiPcs (**42e**, **42f**, **42g**, and **42h**) with strong electron-withdrawing groups significantly reduced the average *V*_T_ to 4.8 V [112], and the *V*_T_ decreased with the increase in the Hammett parameter of the axial substituent groups. However, all the n-type SiPcs OFET devices reported above show poor stability in air. Using the physical vapor deposition method, a thin-film transistor of bis(pentafluorophenoxy) silicon phthalocyanine (F_10_-SiPc) (**42d**) [113] displayed a high electron mobility of 0.54 cm^2^ V^−1^ s^−1^ under a vacuum. However, the devices changed to ambipolar behavior in air, with carriers mobilities about 5 × 10^−3^ cm^2^ V^−1^ s^−1^ for both holes and electrons. Furthermore, perfluorinated silicon phthalocyanine (F_2_-F_16_SiPc) (**43a**) was synthesized and examined [114]. The thin-film transistors based on OTS-modified Si/SiO_2_ demonstrated the highest electron mobilities of 0.30 in N_2_ and 0.17 in air, which are higher than those of F_16_CuPc (**43b**) at the same conditions.

**Table 5 nanomaterials-13-01750-t005:** Representative performance data of phthalocyanine in n-type OFETs.

Material	LUMO (eV)	HOMO (eV)	*µ* (cm^2^ V^−1^ s^−1^)	*I*_on_/*I*_off_	*V*_T_/V	Ref.
**38a**			1.1			[98]
**38b**			0.8			[98]
**38d**			0.12 ± 0.01	1 × 10^5^	+(7 to 25)	[101]
**39a**			(2.0 ± 0.1) × 10^−3^			[102]
**39b**			0.44	1 × 10^3^	37	[103]
**40**			0.3	10^6^		[104]
**41**			(0.02–0.7) × 10^−2^	10^0^–10^4^	24–53	[105]
**42a**			9.25 × 10^−4^	10^2^–10^3^	23.0 ± 1.3	[106]
**42b**			3.28 × 10^−4^	10^2^–10^3^	17.5 ± 2.7	[106]
**42c**			5.56 × 10^−5^	10^1^–10^2^	15.0 ± 0.6	[106]
**42d**	−4.1	−5.77	0.27 ± 0.10		23.5 ± 0.4	[110]
**42e**	−3.9	−5.4	1.3 ± 0.70	10^3^−10^4^	24 ± 1.7	[112]
**42f**	−3.9	−5.4	8.8 ± 3.8	10^5^	24 ± 4.2	[112]
**42g**	−3.9	−5.4	0.90 ± 0.69	10^3^−10^4^	8.1 ± 5.0	[112]
**42h**	−3.9	−5.5	0.52 ± 0.43	10^2^−10^3^	4.8 ± 3.1	[112]
**43a**			0.17	10^5^	11.4 ± 2.1	[114]
**43b**			0.061	10^5^	3.3 ± 3.2	[114]

#### 2.2.2. Perylene Imide-Containing Macrocycles

Perylene imide is one of the most important building blocks of n-type semiconductor materials with high electron mobility [115,116,117] due to its high electron affinity, large intermolecular π-orbital overlap, and ease of functionalization. As mentioned above, macrocyclization is also an effective way to improve the electron mobility of n-type semiconductor materials. Recently, two PDI-containing macrocycles (PDICMs) (cPBPB and cP_4_) (**44**, **45**) (Figure 8) [95] and their linear counterparts were synthesized and characterized by Nuckolls et al. The transistors of **44** and **45** showed the same electron mobilities of (1.5 ± 0.2) × 10^−3^ cm^2^ V^−1^ s^−1^ (Table 6), which were five-fold and about two orders of magnitude higher than those of their linear counterparts. Similarly, the macrocycle **46a**, composed of bithiophenes and PDI and its bromide 1-Br_12_ (**46b**), was also synthesized [118]. However, the mobility of **46b** of about 1.5 × 10^−2^ cm^2^ V^−1^ s^−1^ is far more than that of **46a** (about 6.8 × 10^−4^ cm^2^ V^−1^ s^−1^) due to its robust self-assembly property. Additionally, the macrocycles *cis*-cDBDB (**47**) and *trans*-cDBDB (**48**) [119] were synthesized to investigate the influence of the molecular conformation on electron transport. As a result, the transistor of **47** showed a higher electron mobility (about 4.1 × 10^−3^ cm^2^ V^−1^ s^−1^) than **48** (about 9.9 × 10^−4^ cm^2^ V^−1^ s^−1^). This is because the flexible conformation of **47** is more conducive to charge transport than that of **48**. However, the mobility of **47** of 0.4 × 10^−3^ cm^2^ V^−1^ s^−1^ is lower than that of **48** (1.3 × 10^−3^ cm^2^ V^−1^ s^−1^) [120]. Because the morphology was excluded, the researchers attributed the difference to the higher intramolecular conductivity of *trans*-**44** compared with that of *cis*-**44**. Moreover, by using the cavity of *trans*-**44**, the supramolecular complex ([*trans*-**44**]⸧[PC_61_BM]) [121] exhibited an electron mobility of 8.0 × 10^−3^ cm^2^ V^−1^ s^−1^, which is over five-fold higher than that of neat *trans*-**44**.

## 3. Application of Macrocycles in OLED Devices

Organic light-emitting diode (OLED) devices are electroluminescent devices that generate light emission via the recombination of electrons and holes within an organic layer [122,123,124,125]. Since the first low driving voltage emission from a light-emitting diode based on an organic two-layer structure was reported by Tang and Vanslyke in 1987, OLEDs have attracted much attention from academia and industry. They offer advantages over traditional LED technologies, including low cost, high efficiency, fast response times, light weight, and the ability to produce vivid colors and flexible displays. OLEDs are commonly used in electronic devices, such as smart phones, smart watches, and lighting systems.

Traditional OLED devices are sandwich-type single-layer devices composed of an organic emissive layer (EML) sandwiched between cathode and anode electrodes (Figure 9a). To enhance device performance, various functional layers have been introduced on the basis of the original single-layer device, including electron injection layers (EILs), electron transport layers (ETLs), hole-blocking layers (HBLs), electron-blocking layers (EBLs), hole transport layers (HTLs), and hole injection layers (HILs), ultimately forming a multilayer configuration (Figure 9b). Particularly, introducing an EIL and ETL facilitates electron transport from the cathode to the EML, while the HIL and HTL facilitate hole transport from the anode to the EML. To prevent the carriers from prematurely recombining and to improve device efficiency, the HBL and EBL are incorporated.

In general, several factors can influence the performance of OLED devices, including the device fabrication techniques, device architecture, semiconductor materials, emitters, electrode materials, and gate-insulating materials. The optimization of these factors is essential for achieving high-performance OLED devices with improved efficiency, brightness, and lifetime. In this section, we mainly focus on the structure–property relationship between macrocycle materials and device performance from the perspective of the macrocycle semiconductor materials and emitters used in OLEDs. Compared with linear molecules, macrocycles have several advantages in their application to OLEDs. Firstly, their rigid structure and reduced degrees of freedom lead to stronger intermolecular interactions, resulting in higher thermal and mechanical stability. Secondly, their larger size allows for better energy transfer between molecules, which can lead to higher quantum efficiency and lower operating voltage. Finally, their large conjugated systems provide a larger surface area for charge transport, leading to improved charge injection and transport properties. These advantages make macrocycle molecules promising candidates for use in high-performance OLEDs. In this context, the performance of OLED devices can be significantly enhanced by designing and synthesizing novel macrocyclic organic semiconductor and emitter molecules. Specifically, to improve the film-forming properties and mitigate the luminescence quenching caused by molecular aggregation, it is common to design macrocycles with three-dimensional configurations or introduce spatially bulky groups onto a planar macrocycle skeleton. In the case of phosphorescent OLEDs, the utilization of macrocycles with inherent porous structures coordinated with metals enables the development of second-generation phosphorescent materials with internal quantum efficiencies reaching 100%. Furthermore, employing macrocycles with thermally activated delayed fluorescence (TADF) properties as the emitting layer represents a vital strategy for enhancing the external quantum efficiency (EQE) of OLED devices, as TADF-active organic emitters can theoretically harvest 100% of the excitons generated in the emitting layer through efficient reverse intersystem crossing (rISC) and convert them into light. Typically, the design of macrocycles with TADF properties involves incorporating large π-electronic donor–acceptor (D-A) dihedral angles and vibrational motion within the D-A π-conjugated systems. Finally, the EQEs of representative macrocycles in OLEDs are shown in Figure 10.

### 3.1. Phthalocyanine

Taking advantage of their structural diversity, high thermal and chemical stability, low cost, and appropriate HOMO energy levels, phthalocyanines (**1**) are widely used as HTLs [126], HILs, HBLs, and EMLs in OLEDs. Among them, H_2_Pcs and CuPcs have been the most widely investigated materials mainly due to their early synthesis and their widespread application in the dye industry. In 1991, by using CuPc (**2a**) as the HTL [127], Takeuchi et al. investigated the effects of the crystallinity of the HTL on OLED performance. The crystallinity of **2a** was controlled by changing the temperature of the substrate during the vacuum evaporation of the **2a** layer. It was found that the performance of the OLED with amorphous **2a** as the HTL was better than that of the polycrystalline one in terms of luminosity and uniformity. Although phthalocyanines were used as the HTL in an OLED, the hole transport performance was not as good as that of *N*,*N′*-bis-naphthy-*N*,*N′*-bis-phenyl-benzidine (NPB), which has been widely used and has strong electron-donating properties. However, when a 15 nm **2a** layer was inserted at the interface of the indium tin oxide (ITO) anode and the NPB layer, the aluminum 8-hydroxyquinolinate (Alq3) emitter device showed an half-life time that was increased up to 4000 h because the **2a** HIL decreased the hole injection barrier [128]. Furthermore, to enhance performance, such as power conversion efficiency and operational lifetime, interface modification is usually required at the electrode/organic interface. Therefore, a mixture of isomers of tetra-methyl substituted copper (II) phthalocyanine (CuMePc) (**49a**) (Figure 11) was reported as the HIL [129]. The use of **49a** decreased the crystallization of the HIL and enhanced the film morphology compared with **2a**. As a result, an OLED using **49a** as the HIL exhibited a higher current efficiency (CE = 4.08 cd A^−1^) and operational lifetime (1314 min) than that using a **2a** layer (3.13 cd A^−1^ and 1074 min) and without an HIL (3.30 cd A^−1^ and 18 min) at the optimum thickness of 7.5 nm.

Considering that the strongly acidic poly(styrene sulfonic acid) (PSS) can etch the ITO layer during the spin-coating PEDOT:PSS process and thus affect device stability [130], Xu et al. reported two soluble tetraalkyl-substituted CuPcs (**50a** and **50b**) [131] as anode buffer layers in an Alq3-emitter OLED. The hole-blocking characteristics of Pcs layers markedly impeded hole injection into the EML and thus decreased the production of unstable cationic Alq3 species. Therefore, the OLED stability with **50a** (half-lifetime of 1783 min) and **50b** (2171 min) buffer layers was better than that without a buffer layer (355 min) and than that containing the widely used PEDOT:PSS one (619 min). When **2a** (Table 7) was used as an emitting material, OLED devices usually emitted weak light in the red and near infrared regions [132,133]. One reason for this weak luminescence is that unsubstituted **2a** is prone to aggregation through π–π interaction, leading to remarkable quenching of the emission. Moreover, unsubstituted **2a** is insoluble in common organic solvents, which limits the use of solution processing methods such as spin coating. Hence, a soluble tetra (2-isopropyl-5-methylphenoxyl)-substituted metal-free phthalocyanine (tetra-H_2_Pc) (**51**) [134] was reported as an EML with red and near-infrared (NIR) luminescence. Finally, it was found that the intensity was increased by about fourteen times compared with that of a device based on **1** having the same device structure.

To reduce the use of toxic solvents and molecular aggregation, tetramethyl-substituted aluminum phthalocyanine (AlMePc) (**49b**), which can be stably dispersed in an environmentally friendly mixture of ethanol and butanol via ultrasonication, was reported for use in OLED anode buffer layers [135]. The hole-blocking properties of the **49b** layer produced twice as much luminescence as without the buffer layer and a higher CE of 4.74 cd A^−1^ as well as a power efficiency (PE) of 2.81 lm W^−1^ compared with the diode containing the widely used PEDOT:PSS layer. Moreover, because the film preparation process for **50b** involves the use of a nonaqueous solution, its OLED tolerance is also higher than the 72 min of single HTL devices and the 176 min when inserting a PEDOT:PSS layer device. In addition to **49b**, Lessard et al. reported a variety of axially substituted metal and nonmetallic phthalocyanines (ClAlPc (**52**), Cl_2_-SiPc (**42r**), Cl_2_-GePc (**53a**), Cl_2_-SnPc (**53b**), and F_10_-SiPc (**42d**)) as the HTL produced via vacuum deposition [136]. Among them, the OLED devices containing **42r**, **53a**, and **53b** (with similar physical and electronic properties) showed a similar low maximum luminance. It was worth mentioning that **52** and **42d** provided comparable performance to the well-known hole transport material NPB when using Alq3 as the EML. However, due to the poor solubility of SiPcs in common solvents, it is difficult to use solution processing techniques, such as spin coating, to prepare its devices. To enhance solubility, SiPc **54a** and **54b** disubstituted at the axial positions with carboxylate groups were synthesized and used as emitters to fabricate near-infrared (NIR) OLEDs via spin coating [137]. When the EMLs were doped with 1 wt% in an *N*,*N′*-dicarbazolyl-4-4′-biphenyl (CPB)/2-(4-tert-butylphenyl)-5-(4-biphenylyl)-1,3,4-oxadiazole (PBD) system, the SiPc **54a** and **54b** devices both showed NIR (λ_EL_ = 698–709 nm) electroluminescence properties, and the highest external quantum efficiencies (EQEs) were 0.64% and 1.4%, respectively. Furthermore, inspired by the phenoxylation of tetravalent SiPc, which improves the efficiency of organic solar cell devices by an order of magnitude, Bender et al. used phenoxy-based axial disubstituted **2h**, **42m**, and **42d** as the dopant red emitters and compared the performance of OLED devices prepared by solution and vacuum processing [138]. By using poly(9,9-dioctylfluorene-alt-benzothiadiazole) as the host material, **42m** and **42d** devices fabricated by spin coating exhibited the highest EQEs of 1.4% and 2.5%, respectively. However, the vapor-deposited device made of **42d** with 4,4′,4′′-tris(carbazol-9-yl)-triphenylamine (TCTA) as the host material showed characteristic emission around 715 nm and the highest EQE of 0.09% for 5 wt% doping.

**Table 7 nanomaterials-13-01750-t007:** Representative performance data of phthalocyanine in OLEDs.

Material	HOMO/LUMO (eV)	*V*_on_/DV (V)	CE (cd A^−1^)	PE (lm W^−1^)	EQE (%)	Ref.
**2a**	−5.2/−3.5	3.5/	3.57	1.32		[129]
**42d**	−5.7/−3.8	7.5/			0.15	[138]
**42m**	−5.5/−3.6	4.7/			1.4	[138]
**42r**	−5.8/−3.9	7.0/	<0.1	<0.1		[136]
**49a**	−5.1/−3.4	4.6/	4.08	1.26		[129]
**49b**	−5.3/−3.5	3.3/	4.74	2.81		[135]
**50a**	−5.1/−3.4	4.0/	3.24	1.36		[131]
**50b**	−5.2/−3.4	4.1/	3.26	1.18		[131]
**51**	−5.52/−3.81					[132]
**52**	−5.7/−4.0	2.8/	1.4	1.4	0.32	[136]
**53a**	−5.8/−3.9	4.2/	<0.1	<0.1		[136]
**53b**	−5.8/−3.7	9.0/	<0.1	<0.1		[136]
**54a**	−5.39/−3.64	9.9/			0.64	[137]
**54b**	−5.4/−3.66	7.4/			1.4	[137]

### 3.2. Porphyrin

The macrocycle material porphyrin exhibits high chemical and thermal stability, and its optical properties (absorption and emission) have made it widely used in many applications such as OLEDs. Of course, the narrow emission linewidth of porphyrins also makes them attractive materials in OLEDs. Due to porphyrin possessing a high-oscillation-strength Soret-band absorption in the blue range, the fluorescent red dye tetraphenylporphyrin (TPP) (**15a**) was doped into the blue-light-emitting polymer poly(9,9-dioctylfluorene) (PFO) [139], and saturated red emission was obtained via single-step Förster transfer with a large Förster radius *R*_0_ of 5.4 nm. The OLED device showed an EQE of 0.9%, which is higher than that of PFO LEDs (0.2%) without **15a**. To improve solubility and prevent aggregation, porphyrin functionalized with 9,9-dialkylfluorenyl (**55a**) (Figure 12) was synthesized [140]. By using blended films composed of polyspirobifluorene copolymer host and **55a** (5 wt%) as the EML, the device showed deep-red emission (λ_max_ = 665 nm) with an EQE of 2.5% (Table 8). To reveal the relationship between the structures of porphyrins and the EL properties of OLEDs, six substituted tetraphenylporphin TRPPH_2_ (R = H, CH_3_, OH, F, Cl, and Br) (**55b**–**55g**) compounds were synthesized and used as red EL dopants [141]. The results showed that the more electron-donating the substituents, the more bright and efficient the emission, and the lower the turn-on voltage.

Owing to the lone pair of electrons on pyrrole, porphyrins can coordinate with a metal in their center and enhance the rate of the intersystem crossing between the first singlet and triplet states. Platinum porphyrins have been the most widely investigated for their emission of a saturated red/NIR light. The first example of introducing Pt(II) porphyrin into OLEDs as phosphorescent emitter was based on Pt(II) octaethylporphine (PtOEP) (**15a**) [142]. The doped multilayer EL devices produced a saturated red emission (650 nm) with a peak EQE of 4% using 6% dopant. Moreover, a series of facially encumbered and bulky meso-aryl substituted Pt(II) porphyrins (**56a**, **56b**, **56c**, and **57**) were tested in red-phosphorescent OLEDs as dopants and exhibited a maximum EQE of 1, 4.2, 7.3, and 8.2%, respectively [143]. The trend of increasing EQEs may be related to the facial encumbrance and the steric bulkiness of Pt(II) porphyrins.

To obtain more efficient NIR OLEDs, Thompson et al. developed an efficient phosphorescent emitter by using a nonplanar Pt(II) porphyrin, Pt(II)–tetraphenyltetrabenzoporphyrin [Pt(tpbp)] (**58**), with π extension at the *β*-pyrrole positions [144]. By using Alq3 as the host, the devices displayed NIR (765 nm) emission with a peak EQE of 6.3% at 0.1 mAcm^−2^ and an operational lifetime longer than 1000 h with 90% efficiency at 40 mAcm^−2^. Furthermore, when the 2,9-dimethyl-4,7-diphenyl-1,10-phenanthroline (BCP) was used as the exciton blocking layer (EBL) and ETL, the maximum EQE of the phosphor-**55**-doped devices reached 8.5 ± 0.3% with an emission peak at 772 nm [145]. By further extending the π system attached to pyrrole, Pt(II)tetraphenyltetranaphthoporphyrin (Pt(tptnp)) (**59**) was synthesized by Schanze et al. and was doped into the host 4,4′-bis(carbazol-9-yl)biphenyl (CBP) as the EML [146]. The optimized, multilayer, deposited OLED showed NIR emission (~900 nm) with a peak EQE of ~3.8% and a maximum radiant emittance (R_max_) of 1.8 mWcm^−2^. To achieve an emission wavelength longer than 900 nm, researchers further increased the π system, Pt(II)tetra(3,5-di-tert-butylphenyl)tetraanthroporphyrin (Pt-Ar_4_TAP) (**60**), and obtained a peak electroluminescence wavelength at 1005 nm [147]. Meanwhile, a series of Pt(II)tetrabenzophorphyrins was designed for investigating the structure–property relationships of structural variations and their OLED performance. A Pt-Ar_4_TBP (**61**) device exhibited a peak EQE of 9.2% and an R_max_ of 4.4 mW cm^−2^, which is higher than that of **58**. This may be because the larger number of bulky groups decreased the intermolecular interactions of porphyrin and thus suppressed the triplet–triplet annihilation. Bulkier disubstituted derivatives (Pt-Ar_2_TBP (**62**) and Pt-Ar_2_OPrTBP (**63**)) displayed higher EQEs (7.8% and 6.8%) than Pt-DPTBP (**64**) (5%), while the efficiency of Pt-TAr_2_TBP (**63**) (3.2%) was found to be lower than that of **64**. Moreover, π-conjugated zinc porphyrin hexamers, linear P6 and cyclic c-P6T [148], were also tested via spin-coating in NIR OLEDs as dopants. Compared with P6, c-P6T exhibited further emission redshifting (emission peak at 960 vs. 883 nm) and a higher EQE (0.024 vs. 0.009%) due to the 3D macrocycle structure, which reduces intermolecular aggregation and possesses better π conjugation.

Although the emission wavelength of Pt(II)-porphyrin can be shifted to a deep NIR region by extending the π system of the pyrrole ring, further prolonging the π conjugation leads to a high molecular weight, which affects the evaporation in a vacuum. To solve this problem, one approach involves using the Pt (II) azatetrabenzoporphyrins strategy proposed by Li et al., in which the *meso*-carbon atoms are replaced by nitrogen atoms to modify the porphyrin π system [149]. Pt (II) azatriphenyltetrabenzoporphyrin(PtNTBP) (**66**) and Pt (II) cis-diaza-diphenyltetrabenzoporphyrin (*cis*-PtN_2_TBP) (**67**) devices, fabricated via the thermal vacuum evaporation process, displayed maximum EQEs of 2.8% and 1.5%, respectively, with emission peaks at 848 nm and 846 nm, respectively. This indicated that the emission peak wavelengths of **66** and **67** were redshifted by nearly 70 nm compared with that of **58**. Another method is to introduce eight electron-withdrawing fluorine atoms to the phenyl group of PtTPTNP (**68a**), named Pt(II) tetra(3,5-difluorophenyl)tetranaphthoporphyrin (PtTPTNP-F_8_) (**68b**) [150]. A PtTPTNP-F_8_-based OLED showed a maximum EQE of 1.9% and an EL emission peak at 920 nm. It is worth mentioning that 84% of the emission wavelengths its OLED device, using a stable device structure, were more than 900 nm, and its operational lifetime was longer than 1000 h with 99% efficiency at 20 mAcm^−2^.

Considering the sustainability of the material, a porphyrin derivative meso-tetra[4-(2-(3-n-pentadecylphenoxy)ethoxy]phenyl porphyrin (H_2_P) (**69a**) was synthesized based on cardanol, which was obtained from cashew nut shell liquid, a byproduct of the cashew nut industry [151]. Then, **69a**, copper porphyrin (**69b**) and zinc porphyrin (**69c**) were used as EMLs, and a **69c** device exhibited the strongest emission and the maximum irradiance of 10 µW cm^−2^, while **69b** emitted in the NIR region at 800 nm.

Although phosphorescent OLEDs exhibit higher EQEs than traditional fluorescent OLEDs, the practically useful phosphorescent materials are essentially limited to expensive and toxic rare earth and transition metal complexes [152]. To overcome these shortcomings, Adachi et al. developed the first TADF OLED by using tin(IV) fluoride-porphyrin complexes SnF_2_-OEP (**70**) as the dopant [153]. Although the EL efficiency was very low, some guidelines were determined to enhance the EL efficiency of TADF as follows: (i) the molecular structure should be rigid to reduce thermal deactivation processes; (ii) to promote single and triplet mutual intersystem exchange, the complex should contain heavy atoms, such as metals or halogens; (iii) to increase the rate constant of reverse intersystem-crossing (*κ*_RISC_), the material should possess a small energy gap between the S_1_ and T_1_ states (Δ*E*_ST_).

**Table 8 nanomaterials-13-01750-t008:** Representative performance data of porphyrins in OLEDs.

Material	HOMO/LUMO (eV)	*V*_on_/DV(V)	CE (cd A^−1^)	PE (lm W^−1^)	EQE (%)	Ref.
**55a**		3/			2.5	[140]
**55b**		13.5/		19.7 × 10^−3^		[141]
**55c**		15.5/		19.3 × 10^−3^		[141]
**55d**		14.5/		18.9 × 10^−3^		[141]
**55e**		18.0/		16.8 × 10^−3^		[141]
**55f**		18.8/		15.2 × 10^−3^		[141]
**55g**		19.2/		11.5 × 10^−3^		[141]
**56a**					1	[143]
**56b**					4.2	[143]
**56c**					7.3	[143]
**57**					8.2	[143]
**58**	−4.9/−2.5				8.5 ± 0.3	[145]
**59**		2.2/			3.8	[146]
**61**		2.3/			9.2 ± 0.6	[147]
**62**		2.2/			7.8 ± 0.5	[147]
**63**		2.2/			6.8 ± 0.4	[147]
**64**		2.5/			5.0 ± 0.3	[147]
**65**		2.5/			3.2 ± 0.3	[147]
**66**		8/			2.8	[149]
**67**		8/			1.5	[149]
**68a**					3.8	[150]
**68b**	−4.75/−2.97	5.6/			1.9	[150]
**69a**	−5.2/−3.2	8/				[151]
**69b**	−5.1/−3	4/				[151]
**69c**	−5.2/−3	4/				[151]
**70**		10/			5	[153]

### 3.3. Cyclophane

The large steric hindrance of calixarene is beneficial for improving the molecular solubility and reducing the luminance quenching caused by intermolecular aggregation. Specifically, by using calix[4]arene derivatives as a neutral ligand, lanthanide-complexes-based OLEDs were fabricated via spin-coating with polyvinylcarbazole (PVK) as the host, and the emission peak of lanthanide complexes **71** (Figure 13) and **72** were located at 613 and 547 nm, respectively [154]. To prevent the recrystallization and phase separation of pyrazoline, a steric-functionalized pyrazoline derivative called calix[4]arene-pyrazoline (**73**) was synthesized and utilized as a blue emitter [155]. The optimized device displayed a maximum EQE of 1.52% and pure blue emission (Table 9). Similarly, pyrene-functionalized calix[4]arene **74** and **75** were reported in a simple undoped OLED [156]. The spin-coating-processed device made of pyrene-1,3-*alt*-calix[4]arene **74** exhibited blue emission with a peak EQE of 6.4% (CE of 10.5 cd A^−1^) and a maximum luminous efficiency of 4 lm W^−1^. However, the device made of pyrene-cone-calix[4]arene **75** showed a much lower efficiency, with a maximum CE of 2.5 cd A^−1^. These results suggested that the efficiency of the 1,3-alt-calix[4]arene scaffold in dispersing and suppressing the aggregation of chromophores is far greater than that of the cone-calix[4]arene conformer. Moreover, the supramolecular columnar liquid crystalline material **76**, a chalcone-biphenyl amine molded calixarene derivative [157], was also tested in a blue OLEDs either neat or as a dopant, and the devices showed peak EQEs of 0.09% and 0.71%, respectively.

Owing to their excellent chemical, photochemical, and thermal stability, quinacridone derivatives have been widely applied in OLEDs. However, their strong aggregation significantly decreases their solubility and EL efficiencies. Thus, basket-shaped quinacridone cyclophane (**77**) was reported as a green dopant in OLEDs, which exhibited a CE of 11.2 cd A^−1^ [158].

To solve the problem of aggregation-causing quenching (ACQ) and to enhance the emission efficiency in the solid state, several effective methods have been proposed, such as crystallization-induced emission (CIE, and supramolecular-assembly-induced emission enhancement. However, these methods highly rely on intramolecular motion restriction and the twisted conformational control of organic luminophore. Li et al. reported a macrocyclization-induced emission enhancement strategy that functions by suppressing the nonradiative relaxation process via macrocyclization. By using macrocycle BT-LC (**78**) and monomer BT-M as the dopants in a multilayer OLED device [159], a device made of **78** showed a higher peak EQE (2.82%) than BT-M (1.92%).

Since the appearance of anthracene as the first electroluminescent material, a large number of functional emitters based on anthracene have been reported. However, the application of anthracene derivatives as carrier transport materials in OLED has been rarely explored. Thus, the disilanyl double-pillared bisanthracene (^Si^DPBA) (**79**) was designed and synthesized by Isobe et al. and used as bipolar carrier transport material in phosphorescent OLED [160]. When the **79** was used as ETL, the maximum EQE reached 11.0%, which was higher than that of the device using **79** as HTL with a peak EQE of 3.1%. Considering advantage of bipolar property of ^Si^DPBA, an EQE of 8.7% was obtained by using **79** as both the ETL and the HTL. When **79** was doped into the EML, however, no emission was detected probably because the triplet energy level of 79 (triplet energy (*E_T_*) ∼ 2.3 eV) was lower than the guest Ir(ppy)_3_ (*E_T_* = 2.4 eV). Therefore, the building block of dibenzofuran (*E_T_* = 3.2 eV) was introduced into the synthesis of disilanyl double-pillared bisdibenzofuran (^Si^DPBD(O)) (**80**) [161]. The performance of **80** acting as both a host material and a carrier transport material was tested, and an EQE of 4.6% as the host material, an EQE of 7.2% as ETL were recorded, while a negligible EQE (0.2%) was obtained when **80** was used as HTL.

### 3.4. Fluorene-Containing Macrocycles

Fluorene possesses inherently large exciton binding energies and excellent charge delocalization, making it a crucial building block in the molecular design of light-emitting materials that provide a high quantum yield. Due to the excimer/interchain aggregates and fluorenone defects, however, the luminous efficiency of polyfluorene is significantly decreased, and polyfluorene generates green emission. As end-capping or blending polyfluorenes with triphenylamine (TPA) can effectively suppress green emission, fluorene-containing macrocycles (FCMs) (TPAF_3_)_3_ (**81a**) (Figure 14) and (TPAF_5_)_3_ (**81b**) composed of oligofluorene and TPA moieties were synthesized and used as the hole-transporting emitting layer in a solution-processed blue OLED (Table 10) [162]. Compared with poly(9,9-dioctylfluorene) (PFO), the nonconjugated connection between TPA and oligofluorene raised the HOMO energy levels of **81a** and **81b**, and the steric, rigid, three-dimensional macrocyclic structure effectively inhibited intermolecular aggregation. Therefore, the peak CEs of **81a** and **81b** of 0.63 cd A^−1^ and 0.93 cd A^−1^, respectively, were obtained. Moreover, conjugation-interrupted triphenylamine-fluorene macrocycles are very suitable as a host material in solution-processed phosphorescent OLEDs because of their wide-band gap, high triplet energy, good film formation, and film morphology stability. Triphenylamine-fluorene macrocycle (TPAF)_3_ (**82**) devices, with high triplet energy (*E_T_* = 2.79 eV), exhibited a maximum CE of 22.6 cd A^−1^ and a peak EQE of 5.3%, which were achieved by using iridium(III)bis(4,6-(difluorophenyl)pyridinato-N,C2)picolinate (FIrpic) as the dopant [163].

Based on the macrocycle **82**, diphenylphosphine oxide groups were introduced and attached to TPA to synthesize a bipolar macrocycle (TPA-PO)_3_ (**83**) [164], which was used as the host material doped into FIrpic. A solution-processed blue phosphorescent OLED device was obtained with a peak CE of 19.4 cd A^−1^, a maximum PE of 9.0 lm W^−1^, and a maximum EQE of 8.2%. Furthermore, when the 2,7-di-*tert*-butyl-9H-fluorene on macrocycle **82** was replaced with 4,5-diazafluorene, another host material, the bipolar macrocycle (TPA-DAF)_3_ (**84**) [165], was synthesized to produce a solution-processed blue phosphorescent OLED. Based on the FIrpic dopant device, the maximum CE, maximum PE, and maximum EQE were 10.0 cd A^−1^, 4.8 lm W^−1^, and 4.0%, respectively. When a commercially available host material, 1,1-bis((di-4-tolylamino) phenyl) cyclohexane (TAPC), was doped with **84**, however, the peak CE, PE, and EQE were improved to 17.7 cd A^−1^, 11.9 lm W^−1^, and 7.7%, respectively. Recently, a host macrocycle (Cz-F)_4_ (**85**) with a high *E_T_* (2.82 eV) was prepared via replacing the TPA moiety with *N*-(4-methylphenyl)-carbazole and compared with **82** [166]. Based on the (Cz-F)_4_/TAPC ((Cz-F)4:TPAC = 5:5 in weight) mixed host, a solution-processed blue phosphorescent OLED displayed maximum CE, PE, and EQE values of 18.8 cd A^−1^, 14.8 lm W^−1^, and 8.7%, respectively. Furthermore, a device using a ternary mixed-host system (**85**, TAPC and OXD-7 (1,3-bis(5-(4-(tert-butyl)phenyl)-1,3,4-oxadiazol-2-yl)benzene)]) exhibited a low turn-on voltage (2.8 V) and a low CE roll-off, with a 12% decrease from 100 cd m^−2^ to 5000 cd m^−2^.

Different from the above-mentioned conjugation-interrupted fluorene-based macrocycles, Xie et al. reported a π-conjugated macrocycle [4]cyclo-9,9-dipropyl-2,7-fluorene ([4]CF (**86**)) with a strain energy of 79.8 kcal/mol [167]. Compared with its linear counterpart linear quaterfluorene ([4]LF), **86** showed a significant red shift and a strong green emission in solution, film, and crystal forms. An undoped [4]CF-based OLED device was fabricated via solution-processed procedures, and a maximum CE of 0.83 cd A^−1^ was obtained; this is the first OLED that uses a fluorene-based strained macrocycle as the emitter.

### 3.5. Cyclo-Meta-Phenylenes

Since the cyclo-*meta*-phenylene (CMP) macrocycle was first reported in the 1960s [168,169], its derivatives ([*n*]cyclo-*meta*-phenylenes ([*n*]CMP)) have been continuously synthesized by chemists, such as Isobe et al. Due to the absence of thermally labile or electronically biased substituents on its cyclic hydrocarbon skeleton, the molecule exhibits exceptional thermal stability and serves as a bipolar charge-transporting material in OLED devices. The degenerate π systems present in the strain-free macrocyclic hydrocarbon were found to be advantageous for materials-based applications. In this context, [5]- and [6]CMP (**87a**, **88a**) (Figure 14) were synthesized by Isobe et al. through the one-pot nickel-mediated Yamamoto-type coupling reaction, and their bipolar charge carrier transport properties were investigated as HTLs/ETLs in OLED devices (Table 11) [170]. As a result, these hydrocarbon macrocycles preferred hole transport over electron transport. Concretely, the high performance of [5]- and [6]CMP in HTLs was recorded, with EQEs of 13.9 % and 13.2%, respectively, while their devices in ETLs displayed a lower EQE of approximately 5%. After that, they designed a donor/acceptor conjugated [6]CMP (**89**) via periphery modification [171], and the charge carrier transport property was evaluated in the HTLs/ETLs of emitter-doped phosphorescent OLEDs. Compared with unsubstituted [6]CMP, the device made of **89** in HTLs and ETLs showed an improvement in both values with EQEs of 15.5% and 13.7% and PEs of 22.6 lm W^−1^ and 24.8 lm W^−1^, respectively.

Because [5]- and [6]CMP are bipolar carrier transporters, they have the potential to be further developed as multirole base materials in a single-layer OLEDs. Consequently, a series of [5]- and [6]CMPs with varying substituents have been designed, synthesized, and employed as multirole base materials for single-layer OLEDs. To begin, Isobe et al. initiated the multirole base molecular design of [*n*]CMP macrocycles by conducting a screening study for host materials in a multilayer OLED with Ir(ppy)_3_ acting as the emitter [172]. These **87a** and **88a** devices showed a negligible emitting with EQE of 0.0% and 1.0%, respectively. However, the EQE values were 22.8% for 5Me-[5]CMP (**87b**), 7.3% for 3Me-[6]CMP (**88b**), and 5.3% for 6Me-[6]CMP (**88e**), signifying that these materials are suitable as host materials. Subsequently, these *m*Me-[*n*]CMPs were evaluated in a single-layer OLED with 6 wt% Ir(ppy)_3_ as the doped emitter. The EQE values were 5.3% and 7.3% for **88c** and **88b**, respectively. However, the device made of **87b** exhibited a high EQE of 22.8%, which is comparable to that of most multilayer OLEDs. This implied that a donor–acceptor design is not necessary for achieving high efficiency in a single-layer OLED. Additionally, these findings indicated that the wide bandgap motif of aromatic hydrocarbons makes them optimal as a multirole, single-component base material. Additionally, a white-light-emitting single-layer OLED was obtained with an EQE of 10.4%.

Considering the effect of the steric design at the periphery on the performance of single-layer OLED devices, researchers proceeded to design and synthesize 2′-tolyl and 2′-*m*-xylyl substituted arylated [5]- and [6]CMPs (**87c**, **87d**, **88d**, and **88e**) [173]. To evaluate the performance of these aromatic hydrocarbon macrocycles, a single-layer architecture device with two regions was utilized, and 6 wt% Ir(ppy)_3_ was used as the doped emitter. The results showed that EQE values of 24.8%, 18.7%, 14.2%, and 14.6% were obtained for **87c**, **87d**, **88d**, and **88e**, respectively. Moreover, the highest PE (54.4 lm W^−1^) and CE (88.0 cd A^−1^) were recorded for **87c**. These observations validated the significance of the steric design at the periphery of macrocycles.

To accommodate blue Ir emitters (FIrpic) with high *E_T_* levels, trifluoromethylated [*n*]CMPs (*n*CF_3_-[*n*]CMP) were synthesized and utilized as host materials in blue phosphorescent multilayer OLEDs [174]. The EQE of 5CF_3_-[5]CMP (**87e**) (9.9%) was found to be higher than that of 6CF_3_-[6]CMP (**88f**) (6.3%), despite their nearly identical *E_T_* levels (2.67 eV and 2.68 eV, respectively), which cannot explain their differences in performance. A crystal packing analysis revealed that CH-π/CF-π interactions dominated the crystalline solid state of **84e**, facilitating the favorable entrapment of the emitter and resulting in improved charge and energy transfer.

### 3.6. Arylamine- and Triazine-Containing Macrocycles

Since Adachi demonstrated the first TADF-OLED using a SnF_2_-OEP emitter in 2009, a large number of TADF materials have been synthesized and utilized in the field of OLEDs. However, there have been few reports on TADF materials based on macrocyclic structures, especially those used for TADF-OLEDs. To date, the reported macrocyclic TADF materials for OLEDs are mainly arylamine-containing macrocycles (AACMs) [175,176,177] and triazine-containing macrocycles (TRCMs) [178,179,180]. Specifically, Minakata et al. first reported the EQEs of OLEDs fabricated with a macrocyclic TADF emitter: a purely organic macrocyclic D–A–D–A π-conjugated macrocycle **90a** (Figure 15) [175]. The devices based on the TADF emitter macrocycle **90a** and the linear analogue were examined and found to have EQEs of 11.6% and 6.9% (Table 12), respectively. Comparative analysis of the physicochemical properties between the macrocycle and its linear analogue indicated that macrocyclization can be an effective approach to enhance TADF efficiency by suppressing nonradiative pathways. Furthermore, a macrocyclic *t*-BuMC (**90b**) with TADF properties was obtained by introducing *t*-Bu groups onto the TADF macrocycle **90a**, resulting in a TADF-OLED device with an EQE of 10% [176].

Regarding triazine-containing macrocycles, Yasuda et al. reported a π-conjugated TADF macrocycle, MC-C3T3 (**91**), which incorporates both electron-donor and -acceptor units [177]. Macrocycle **91**(Δ*E*_ST_ = 0.13 eV) demonstrated a lower Δ*E*_ST_ than the acyclic fragment molecule (Δ*E*_ST_ = 0.24 eV). The EL characteristics of cyclic **91** and the acyclic fragment molecule revealed that the maximum EQE of **91** was 15.7%, whereas that of the acyclic fragment molecule was only 4.2%. Additionally, a chiral TADF macrocycle with a pair of macrocyclic enantiomers, (+)-(*R*,*R*)-MC (**92**) and (−)-(*S*,*S*)-MC (**93**), was synthesized through the combination of a chiral octahydro-binaphthol moiety with a triazine-based TADF skeleton [180]. The chiral macrocycles exhibited obvious TADF properties with a low Δ*E*_ST_ of 0.067 eV and aggregation-induced emission behavior. The circularly polarized OLED (CP-OLED) displayed two types of chiral macrocycles, with the highest EQEs of 17.1% and 15.5%, and demonstrated circularly polarized electroluminescence properties with a |*g*_EL_| of 1.7 × 10^−3^.

## 4. Application of Macrocycles in OPV Devices

Organic photovoltaics (OPVs), also known as organic solar cells (OSCs), represent a novel class of solar cells that utilize organic semiconductor materials as the active layers (Figure 16). These OSCs have garnered significant attention as promising next-generation photovoltaic technology owing to their relatively straightforward fabrication process, abundant material sources, cost-effectiveness, vibrant coloration, transparency, customizable molecular design, and potential for large-scale roll-to-roll production. The active layers of OSCs play a crucial role in the energy conversion process and typically comprise a blend of organic electron-donor and -acceptor molecules. The photovoltaic performance of these devices hinges upon the selection of suitable organic electron-donor and -acceptor materials. In this context, we explored the notable macrocyclic donor and acceptor molecules that have demonstrated exceptional photovoltaic performance in recent years. Finally, the PCEs of the representative macrocycles in OPVs are shown in Figure 17.

### 4.1. Electron Donor Macrocycles

#### 4.1.1. Porphyrin

Porphyrins and their derivatives have many excellent properties in organic photovoltaics [13,14]. Firstly, porphyrins and their derivatives have a wide absorption range and a high extinction coefficient, which can improve the efficiency of solar energy utilization in battery devices. Secondly, introducing different functional groups on the *meso* or *β* position or modifying metal ions in cavities can adjust the absorption, energy levels, molecular planarity, and molecular stacking properties of small molecules. Thirdly, the large conjugated plane of porphyrin materials promotes *π–π* stacking and charge transfer between molecules. Fourthly, the good chemical stability of porphyrins is conducive to the service life of devices in various environments. Therefore, a series of porphyrin molecules have been developed as donors in OPVs.

In 2022, So et al. reported the application of palladium(II) and platinum(II) porphyrins (**94a** and **94b**) (Figure 18) as electron donors in organic photovoltaics. The devices based on palladium(II) and platinum(II) porphyrin donors showed power conversion efficiencies (PCEs) of 8.09 and 7.31% under 1 Sun (Table 13), respectively, being higher than the efficiency of porphyrin donors without metal. The improvement in the performance of the devices is mainly due to a more effective charge collection, faster charge transfer, more suppressed recombination behavior, and smoother surface morphology [181]. In the same year, Wang et al. used panchromatic terthiophenyl-benzodithiophene conjugated porphyrin (**95**) as the electron donor for an efficient OSC application. The successful binding of trithiophenyl benzodithiophene to the porphyrin center resulted in a narrow optical band gap within 1.58–1.60 eV owing to filling the absorption gap between the Soret and Q bands of traditional porphyrin molecules. The device achieved a short circle current (*J*_sc_) of 14.91 mA cm^−2^, a fill factor (FF) of 66.5%, and a PCE of 8.59% [182].

#### 4.1.2. Phthalocynine

Phthalocyanines, as good organic semiconductor materials, have rapidly been developed as electron donors in OPVs due to their large planar structure, high molar extinction coefficient, and strong stability [183]. In 2009, Nakamura et al. obtained a three-layer p-i-n OPV device composed of a tetrabenzoporphyrin (**16a**) donor and a silylmethyl[60]fullerene acceptor via a new solution-processable fabrication process. The device showed a respectable PCE of 5.2% [184]. In 2011, Heremans et al. used lead phthalocyanine (**2i**) as an electron donor and controlled the structural evolution of lead phthalocyanine (**2i**) thin films by optimizing the deposition conditions for high-performance NIR-sensitive solar cells, thus obtaining a PCE of 2.6% and an EQE above 11% from 320 to 990 nm with a peak value of over 34% at λ = 900 nm [185].

### 4.2. Electron Acceptor Macrocycles

#### 4.2.1. Porphyrin

The excellent performance of devices based on porphyrin donor materials has also inspired researchers to develop porphyrins as electron acceptor materials for OSCs. In 2011, Heeger et al. designed and synthesized a porphyrin-fullerene dyad (**96**) electron acceptor with a supramolecular “double-cable” structure to serve as a new electron acceptor for bulk heterojunction polymer solar cells (PSCs). The PCE of this OSC device was 3.35%, which is lower than the efficiency of fullerene systems under the same conditions. The double-cable structure showed a higher *J*_sc_ and a larger open circuit voltage (*V*_oc_) than that of PCBM, which indicated that the double-cable structure is a promising electron acceptor for high-performance PSCs [186]. In 2014, Therien et al. synthesized a small molecule acceptor material by connecting two isoindigo groups to the ends of porphyrins through an acetylene bridge (**95a** and **95b**). This is the first instance where a single, small porphyrin molecule, typically employed as an electron donor in OPVs, was successfully utilized as a nonfullerene acceptor in solution-processed organic solar cells [187].

#### 4.2.2. Phthalocynine

The development of Phthalocynine as an electron acceptor material in OPVs has also made great progress owing to its good properties and high chemical, thermal, and light stability. In 2021, Lessard et al. used variance-resistant PTB7 and axially substituted silicon phthalocyanine (**41f**) as the electron acceptor for high-*V*_oc_ organic photovoltaics. When the fullerene acceptor was substituted with **41f**, 80% of the overall device efficiency was maintained, which was accompanied by a high average *V*_oc_ of 1.05 V [188]. Furthermore, in 2023, Lessard et al. prepared a flexible, large-area (1 cm^2^) organic photovoltaics device based on low-cost silicon phthalocyanine (**41f**) as a nonfullerene acceptor. The PCE of this device was 1.3% [189].

#### 4.2.3. Other Macrocycles

Perylene imide (PDI) is also used in solar cells due to its high thermal stability and high fluorescence quantum yield. In 2016, Nuckolls et al. built two conjugated macrocycles, **44** and **45,** for solar cells. Macrocycle **44** alternates two electron-donating bithiophene subunits with two electron-accepting diphenyl-perylenediimide subunits. Macrocycle **45** is composed of four electron-accepting diphenyl-perylenediimide units. This is the first time that a macrocycle was used as an electron acceptor in an OPV. The average PCE of **44** was 3.3%, with a maximum value of 3.5%. The average PCE of **45** was 3.5%, with a maximum value of 3.6%, which is higher than that of acyclic molecules because the cyclic acceptors have enhanced photocarrier generation and better charge transport [95]. In 2019, Li et al. used a new macrocyclic molecule (**98**) containing four electron-deficient diketopyrrolopyrrole (DPP) units alternated with electron-rich thiophenes as the electron acceptor for nonfullerene organic solar cells. The device produced a PCE of 0.49%, which is much higher than the efficiency of linear molecules with a similar backbone [86].

**Table 13 nanomaterials-13-01750-t013:** Representative performance data of macrocycles in OPVs.

Material	Function	HOMO/LUMO (eV)	*J*_sc_ (mA cm^−2^)	*V*_oc_ (V)	FF (%)	PCE (%)	Ref.
**94a**	donor	−5.65/−3.88	12.82 ± 0.12	0.94 ± 0.003	67.0 ± 1.28	8.07 ± 0.03 (8.09)	[181]
**94b**	donor	−5.71/−3.86	11.18 ± 0.02	0.98 ± 0.004	66.1 ± 0.53	7.26 ± 0.07 (7.31)	[181]
**94c**	donor	−5.55/−3.90	7.03 ± 0.18	0.67 ± 0.01	51.2 ± 1.01	2.42 ± 0.07 (2.51)	[181]
**95**	donor		14.91	0.866	66.5	8.59	[183]
**16a**	donor		10.5	0.75	0.65	5.2	[184]
**2i**	donor		11	0.47		2.6	[185]
**96**	acceptor		11.5	0.56	0.56	4.03	[186]
**97**	acceptor		2.43	0.79	0.29	0.57	[187]
**41f**	acceptor		−4.6	0.66	0.42	1.3	[189]
**44**	acceptor	−5.39/−3.87	9.2 ± 0.3	0.84 ± 0.01	0.44 ± 0.01	3.3 ± 0.2 (3.5)	[95]
**45**	acceptor	−5.69/−3.90	9.7 ± 0.2	0.83 ± 0.01	0.44 ± 0.01	3.5 ± 0.1 (3.6)	[95]
**98**	acceptor	−4.80/−3.19	1.65	0.67	0.44	0.49	[86]

## 5. Application of Macrocycles in DSSC Devices

Dye-sensitized solar cells (DSSCs) are thin-film photovoltaic cells that mimic the process of photosynthesis in plants. These cells utilize pigment molecules to efficiently transfer energy and electrons, converting visible light into electrical energy. DSSCs have received significant attention as a promising photovoltaic technology due to their ease of fabrication, availability of raw materials, tunability, short energy payback time, and environmental friendliness [190,191]. Sensitizers play a critical role in the light-harvesting capability and energy conversion efficiency of DSSCs, exerting a significant impact on battery performance. Inspired by the solar energy collection mechanism in plant photosynthesis, scientists have designed and synthesized numerous sensitizers. Porphyrins and phthalocyanines, as macrocyclic molecules, employ substituents to shift the absorption bands (Soret and Q bands) to lower energy levels, thereby enhancing the power conversion efficiency (PCE) of the devices [192]. Consequently, porphyrins and phthalocyanines have emerged as the most widely adopted sensitizers. Finally, the PCEs of representative macrocycles in DSSCs are shown in Figure 19.

### 5.1. Porphyrin

In plant photosynthesis, the chromogenic groups of porphyrins capture solar energy and effectively convert the captured radiant energy into chemical energy [193]. Inspired by the efficient energy transfer of photosynthesis, many different porphyrins have been designed and synthesized for DSSCs [15,194,195]. Porphyrin-based dyes have large absorption coefficients in the visible-light region due to their rigid structure. Porphyrin-based dyes have four meso- and eight *β*-site reaction sites, which can perform functional modifications, including the fine tuning of optical, physical, electrochemical, and photovoltaic properties [16]. In particular, a device with a zinc porphyrin sensitizer cosensitized with an organic dye using a cobalt-based electrolyte (**99**) (Figure 20) achieved a PCE of 12.3% (Table 14) [196], which is superior to that of a device based on a Ru complex [197,198]. After decades of development, porphyrin-based (**100**) DSSCs have achieved PCEs exceeding 13%, indicating that porphyrins are promising candidates for the preparation of highly efficient DSSCs [17]. In 2020, Kim et al. used thieno[3,2-b]indole-based organic dyes (**101**) and porphyrin as cosensitizers. The device exhibited a state-of-the-art PCE up to 14.2%, which is higher than that of thieno[3,2-b]indole-based organic dyes and porphyrin as the cosensitizer as sensitizer, respectively [199].

### 5.2. Phthalocynine

Phthalocyanines (Pcs) have great potential in the application of DSSCs owing to their high chemical, thermal, and light stability [200]. Good efficiency has been achieved by using Pcs as the light harvester. To improve the light capture ability of dyes, chromogenic groups have been introduced as secondary energy collectors to supplement the absorption of Pc. In 2012, Zhang et al. synthesized an unsymmetrical anthraquinone-Pc (APC) decorated with three peripheral tert-butoxy groups (**102**) for DSSC applications. The agglomeration of Pc led to the device generating only a modest PCE of 0.71%, but the photoresponse of the Pc-sensitized film was extended up to 750 nm [201]. To construct nonaggregated and low-bandgap Pcs, Zhang et al. implemented a bisethylamino, bispropylamino, or bisbutylamino group around the periphery of the PC. In addition, amino acid moieties were used as anchoring groups. When these Pcs were adsorbed onto a TiO_2_ film, the absorption characteristics were significantly widened and redshifted toward the NIR region. The TPC dye (**103**) achieved a highest photocurrent of 2.07 mA cm^−2^ and a PCE of 1.67% [202].

**Table 14 nanomaterials-13-01750-t014:** Representative performance data of macrocycles in DSSCs.

Material	HOMO/LUMO (eV)	*J*_sc_ (mA cm^−2^)	*V*_oc_ (V)	FF (%)	PCE (%)	Ref.
**99**		17.66	0.935	0.74	12.3	[196]
**100**		20.86 ± 0.2	0.912 ± 0.007	73.2 ± 0.2	13.9 ± 0.3	[17]
**102**	0.90/−0.92	2.04	0.52	0.67	0.71	[201]
**103**	0.90/−0.9	2.07	0.55	0.59	1.67	[202]

## 6. Conclusions and Perspectives

In this article, we mainly focused on the impact of macrocyclic structures on device performance. The key factors that influence the device performance of macrocycles are their energy level structure, structural stability, film-forming property, skeleton rigidity, inherent pore structure, spatial hindrance, exclusion of perturbing end-effects, macrocycle-size-dependent effects, and fullerene-like charge transport characteristics. Specifically, the structure–performance relationships of macrocycles in their optoelectronic devices are manifested in the following aspects: (i) Their rigid structure and limited degrees of freedom result in stronger intermolecular interactions, leading to higher thermal and mechanical stability. (ii) Macrocyclic organic semiconductor molecules with large π-conjugated systems and significant intermolecular π–π overlap are crucial for achieving high carrier mobility. (iii) The energy level structure of macrocycles directly affects the injection and transport of charge carriers, thereby influencing device performance. (iv) Macrocyclization significantly reduces the molecular reorganization energy, thereby enhancing field-effect mobilities. (v) By utilizing the inherent porous structure of macrocycles coordinated with metals, second-generation phosphorescent materials with internal quantum efficiencies reaching 100% can be formed. (vi) The design of macrocycles with three-dimensional configurations or planar macrocycles modified with bulky groups effectively improves the film-forming properties and suppresses the aggregation-induced luminescence quenching, thereby enhancing device performance. (vii) The robust backbone and intramolecular interactions of macrocycles inhibit nonradiative relaxation, leading to improved luminous efficiency in OLEDs. (viii) Macrocyclization can enhance the PCE of OPVs by promoting the generation of photogenerated carriers and enhancing carrier transport. (ix) Compared with the preferences of OFET, OPV, and DSSC devices for fully conjugated macrocycles, macrocyclic structures can be either fully conjugated or conjugation-disrupted in OLEDs.

By designing new macrocyclic structures or modifying existing ones with substituents, the structure–property relationships between macrocycles and their device performance have been continuously revealed and validated, providing a research foundation for obtaining higher device performance in the future. In fact, in addition to the aforementioned applications, macrocycles have also been reported in other optoelectronic devices such as perovskite solar cells [203], organic field-effect transistor memory [204], organic field-effect transistor sensors [205,206], organic photodetectors [85], and memristors [207]. Going forward, macrocycles are expected to find even more applications in the field of optoelectronics, with flexible electronic devices being a key trend in future development. Additionally, our group has recently designed and synthesized a novel macrocyclic structure, organic nanogridarene [208,209,210,211,212,213], which exhibits excellent optoelectronic properties and has demonstrated significant potential for future research in the field of organic optoelectronic devices.

## Figures and Tables

**Figure 1 nanomaterials-13-01750-f001:**
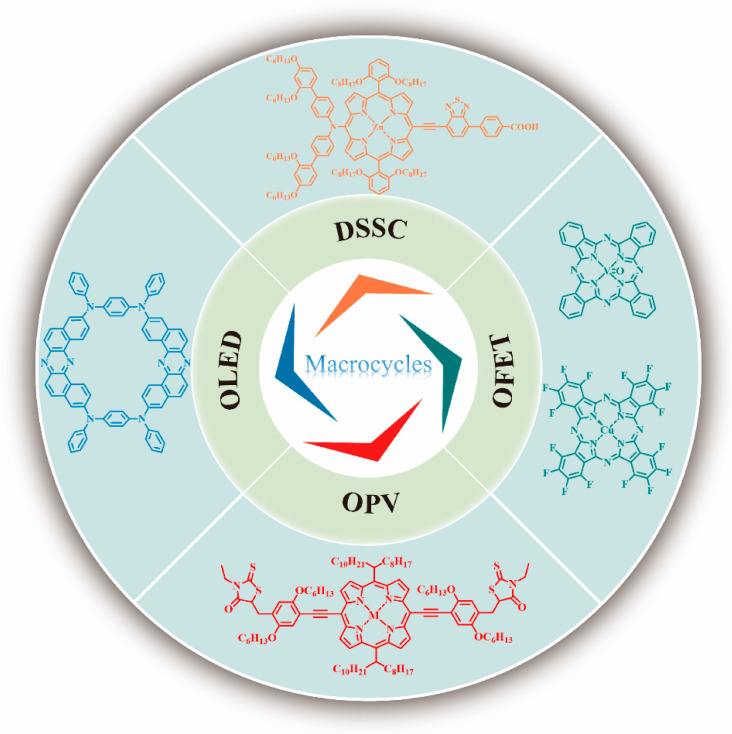
Applications of macrocycles in organic optoelectronic devices.

**Figure 2 nanomaterials-13-01750-f002:**
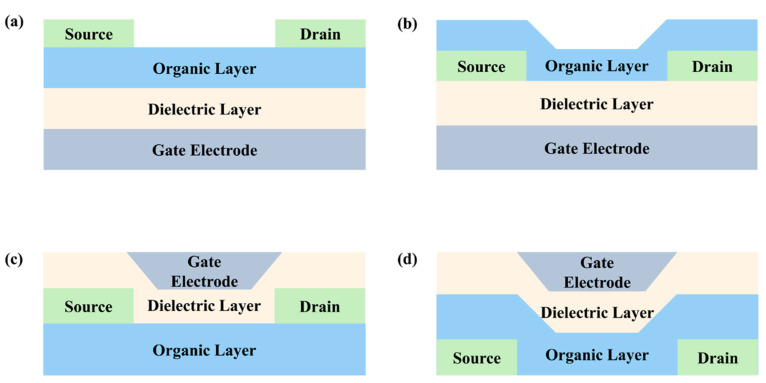
Four types of OFETs: (**a**) bottom gate/top contact, (**b**) bottom gate/bottom contact, (**c**) top gate/top contact, and (**d**) top gate/bottom contact.

**Figure 3 nanomaterials-13-01750-f003:**
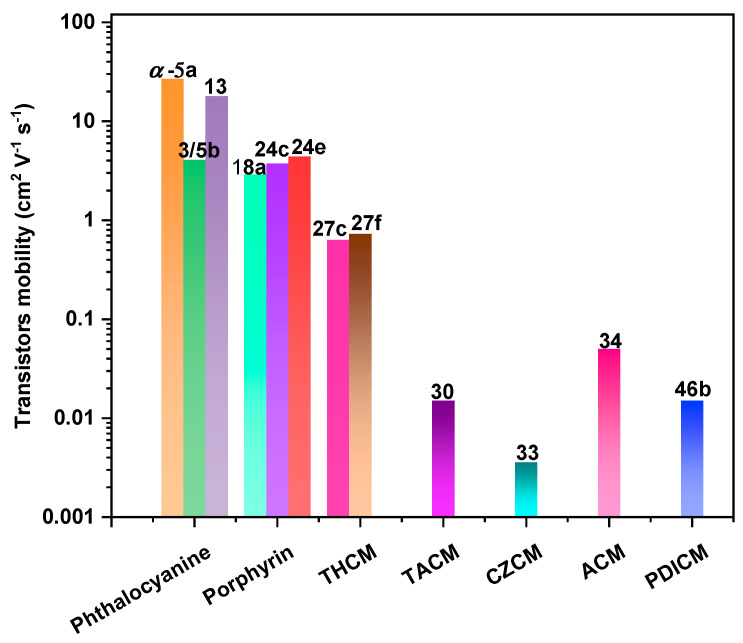
Charge carrier mobilities of representative macrocycles in OFETs.

**Figure 4 nanomaterials-13-01750-f004:**
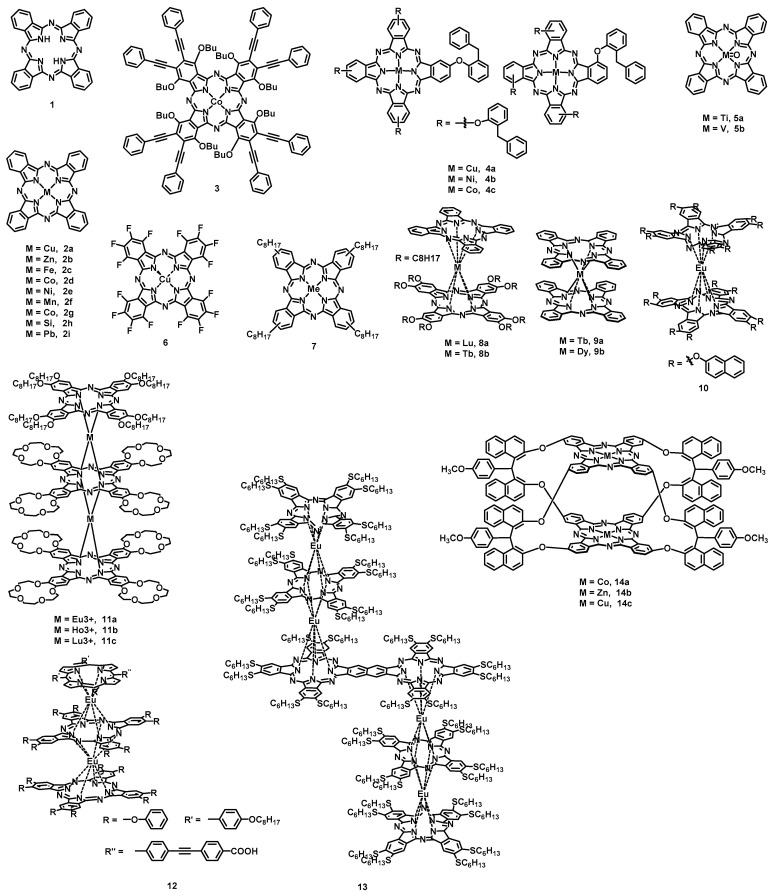
Phthalocyanine and derivatives in p-type OFETs.

**Figure 5 nanomaterials-13-01750-f005:**
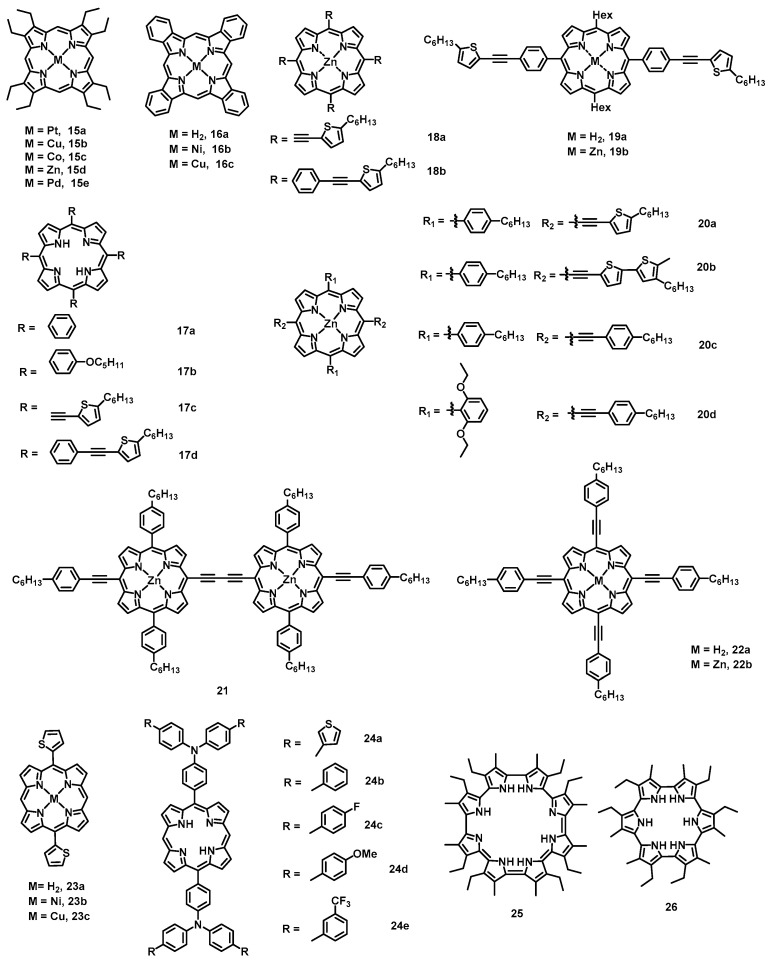
Porphyrin and porphyrin analogues in p-type OFETs.

**Figure 6 nanomaterials-13-01750-f006:**
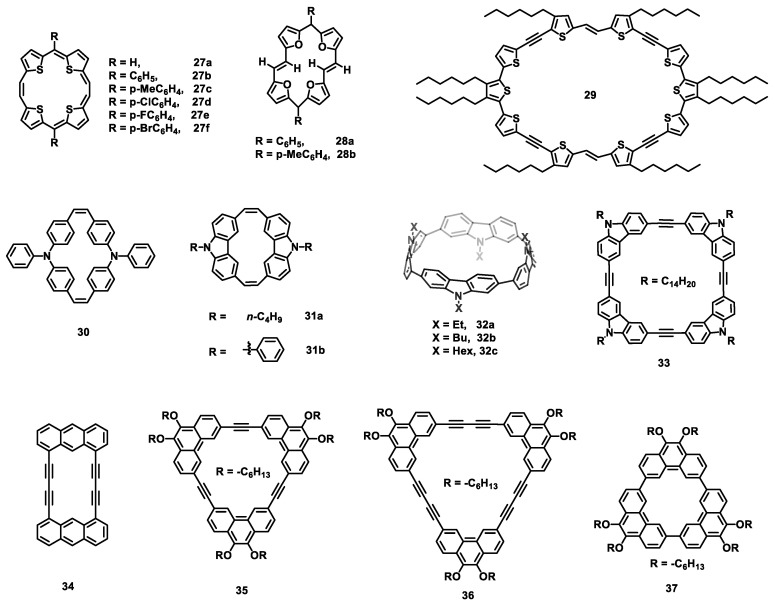
Thiophene-, furan-, triarylamine-, carbazole-, and acene-containing macrocycles in p-type OFETs.

**Figure 7 nanomaterials-13-01750-f007:**
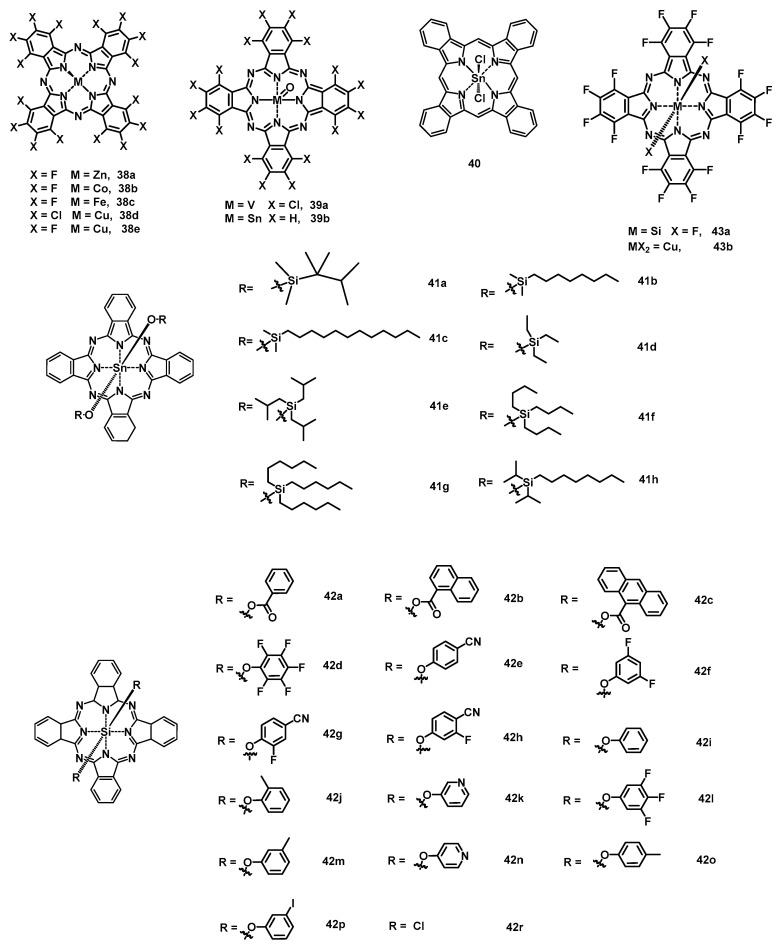
Phthalocyanine and derivatives in n-type OFETs.

**Figure 8 nanomaterials-13-01750-f008:**
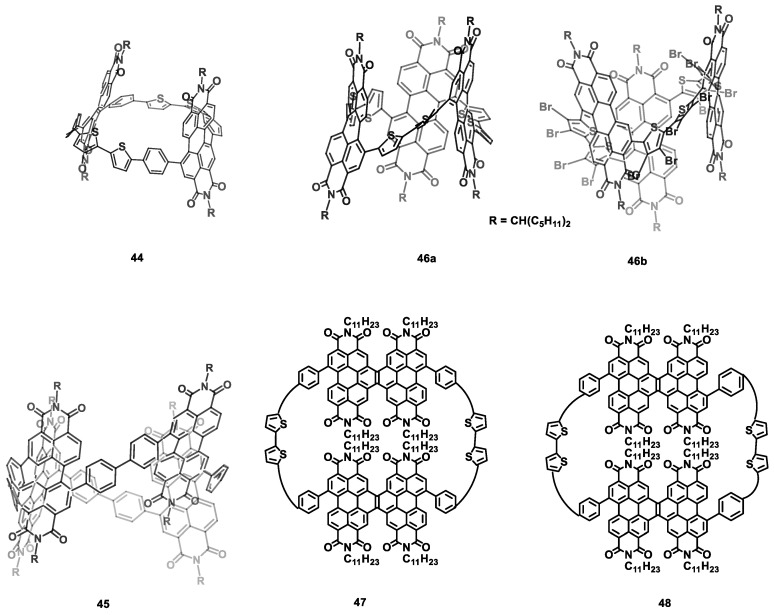
Perylene imide-containing macrocycles in n-type OFETs.

**Figure 9 nanomaterials-13-01750-f009:**
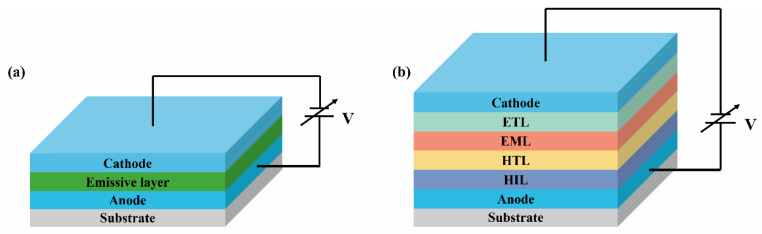
Schematic circuit diagram (**a**) for single-layer OLED; (**b**) multilayer OLED.

**Figure 10 nanomaterials-13-01750-f010:**
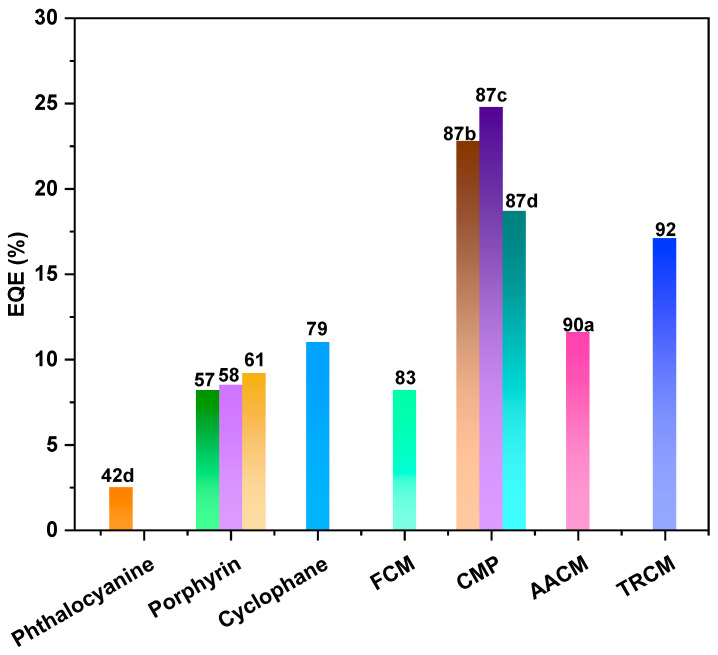
EQE of representative macrocycles in OLEDs.

**Figure 11 nanomaterials-13-01750-f011:**
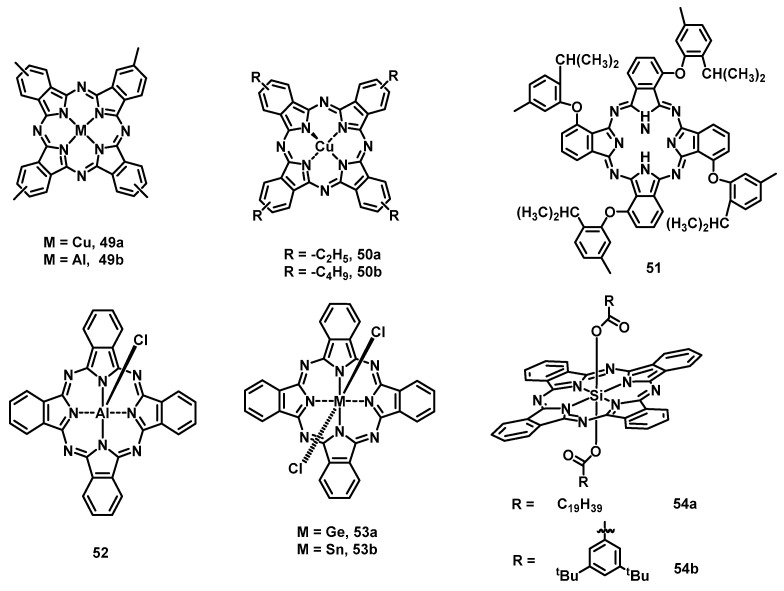
Phthalocyanine and derivatives in OLEDs.

**Figure 12 nanomaterials-13-01750-f012:**
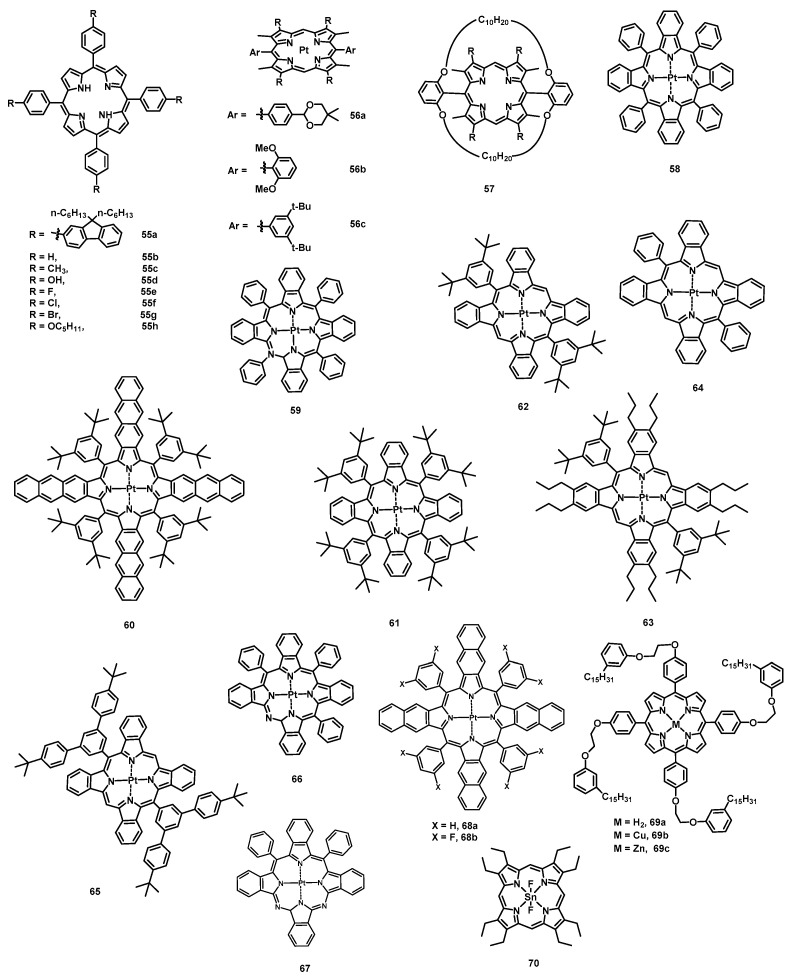
Porphyrin and derivatives in OLEDs.

**Figure 13 nanomaterials-13-01750-f013:**
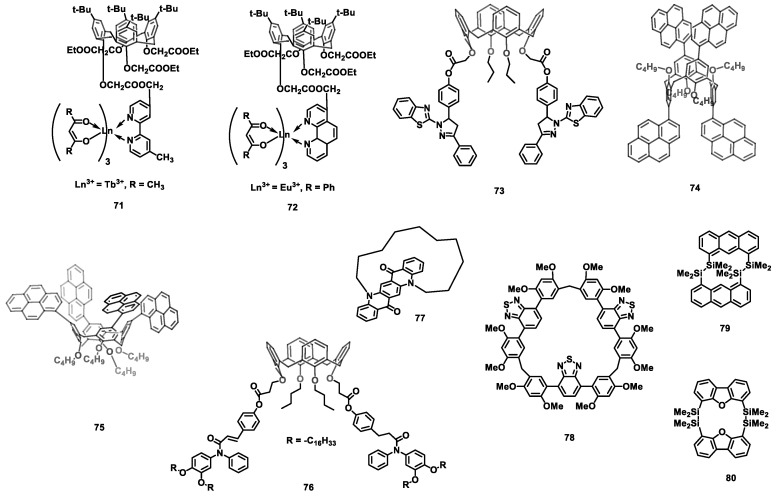
Cyclophane in OLEDs.

**Figure 14 nanomaterials-13-01750-f014:**
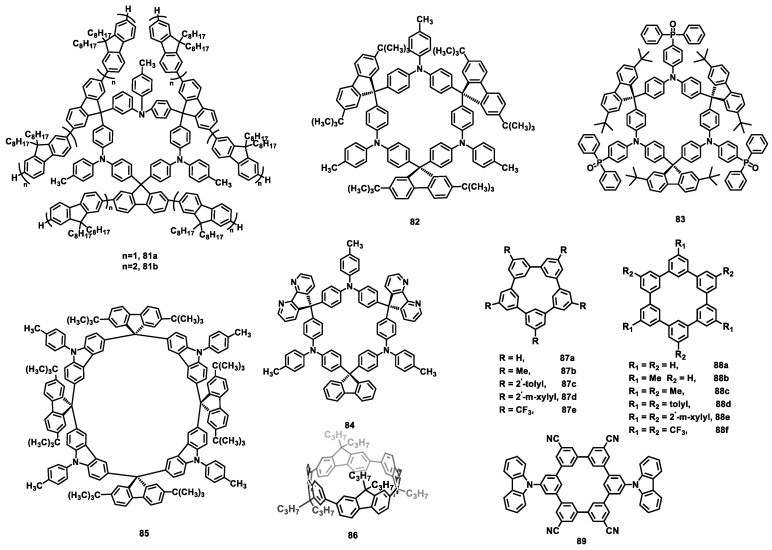
Fluorene-containing macrocycles and cyclo-meta-phenylenes macrocycles in OLEDs.

**Figure 15 nanomaterials-13-01750-f015:**
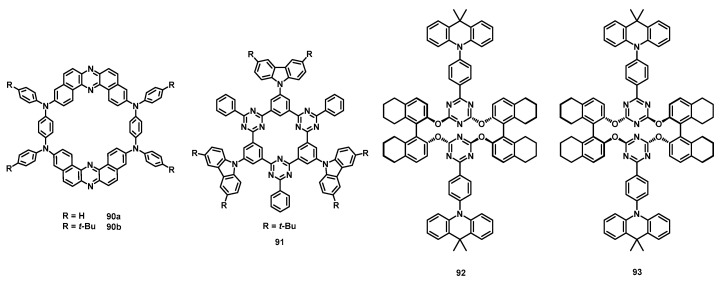
Arylamine- and triazine-containing macrocycles in OLED.

**Figure 16 nanomaterials-13-01750-f016:**
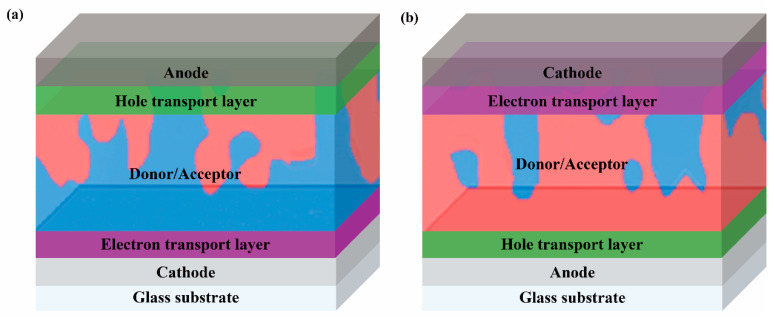
Conventional (**a**) and inverted (**b**) architectures are utilized in bulk heterojunction organic solar cells, where the red regions represent the electron donors, and the blue regions represent the electron acceptors.

**Figure 17 nanomaterials-13-01750-f017:**
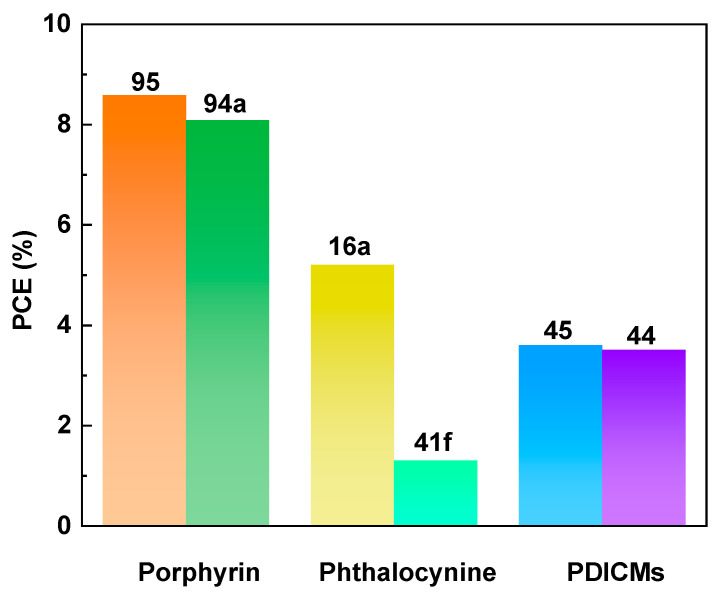
PCEs of representative macrocycles in OPVs.

**Figure 18 nanomaterials-13-01750-f018:**
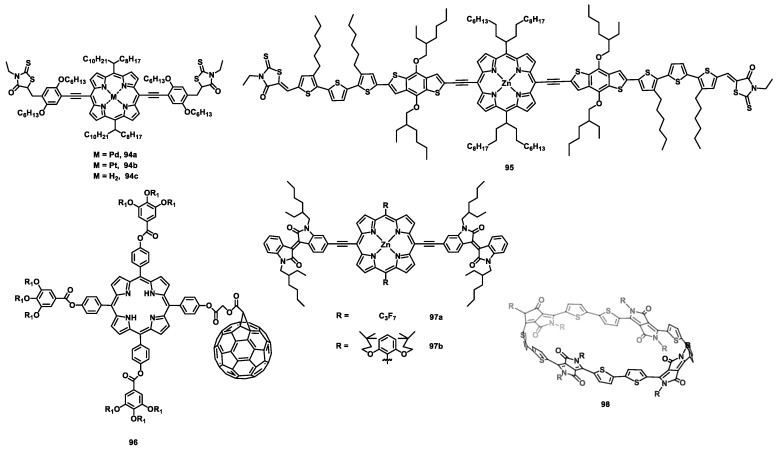
Macrocycles in OPVs.

**Figure 19 nanomaterials-13-01750-f019:**
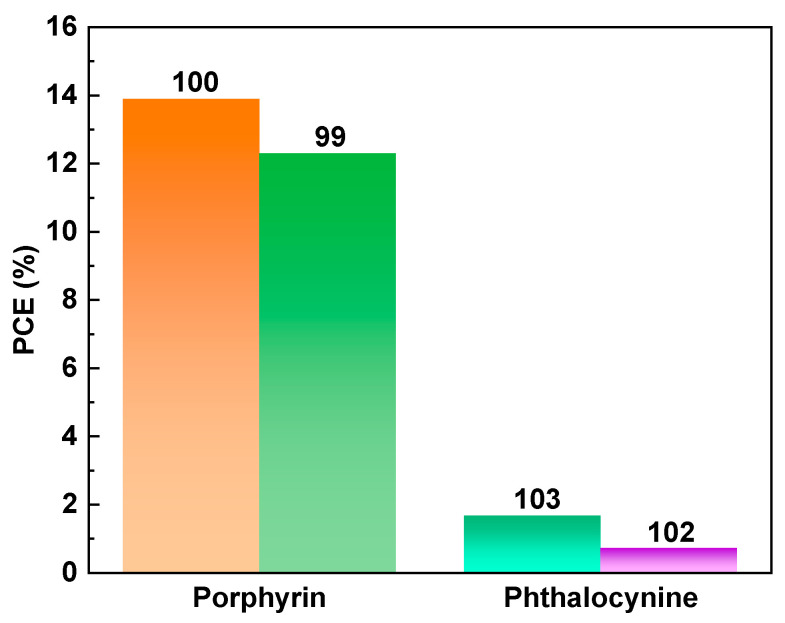
PCEs of representative macrocycles in DSSCs.

**Figure 20 nanomaterials-13-01750-f020:**
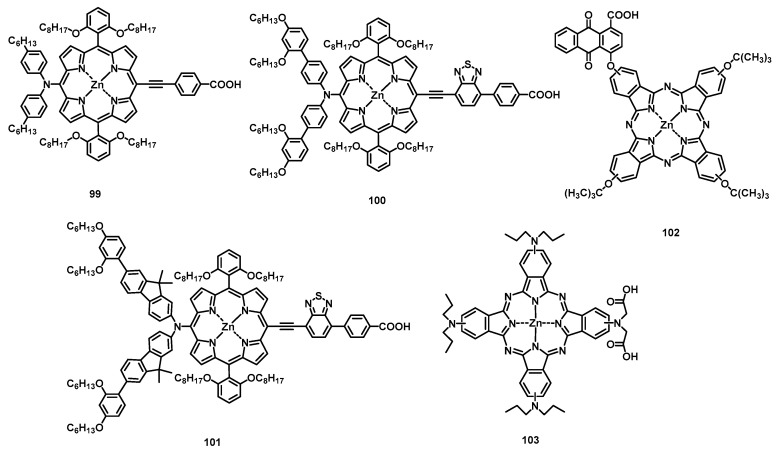
Macrocycles in DSSCs.

**Table 2 nanomaterials-13-01750-t002:** Representative performance data of porphyrins and porphyrin analogues in p-type OFETs.

Material	LUMO (eV)	HOMO (eV)	*µ* (cm^2^ V^−1^ s^−1^)	*I*_on_/*I*_off_	*V*_T_/V	Ref.
**15a**			1.3 × 10^−4^	10^4^–10^5^		[58]
**15b**			0.2			[59]
**15c**			0.068			[59]
**15d**			0.036			[59]
**15e**			0.014			[59]
**16a**			0.017	10^5^	−3.4	[60]
**16b**			0.2	10^3^	−13	[63]
**16c**			0.1	10^4^	5	[64]
**17a**			0.012		−7.5	[65]
**17b**	−3.71	−5.77	1.8 × 10^−3^		−14.5	[66]
**17c**			0.85	10^4^		[68]
**17d**			6.2 × 10^−2^			[67]
**18a**			2.90	6 × 10^3^	2.0	[68]
**18b**	−3.48	−5.28	0.32	10^4^		[67]
**19a**	−3.42	−5.32	0.12	10^6^		[69]
**19b**	−3.54	−5.36	0.36	2 × 10^3^		[69]
**20a**	−3.43	−5.23	0.27	2.2 × 10^3^	−2	[70]
**20b**	−3.37	−5.12	0.066		−8	[70]
**20c**	−3.38	−5.34	2.57	1 × 10^5^	−5.0	[71]
**20d**	−3.58	−5.37	0.48	3 × 10^3^	−6.0	[71]
**21**			0.026	10^5^	−0.60	[72]
**22a**			0.3	10^4^	−12.6	[73]
**22b**			0.2	10^5^	3.6	[73]
**23a**	−2.63	−5.46	0.15			[74]
**23b**	−2.51	−5.70	1.50			[74]
**23c**	−2.55	−5.66	0.74			[74]
**24a**	−2.32	−4.88	0.66	10^8^	−8.6	[75]
**24b**	−2.29	−4.75	0.25	10^7^	−9.4	[75]
**24c**	−2.37	−4.97	3.74	10^8^	−12.2	[75]
**24d**	−2.27	−4.69	0.72	10^6^	−9.0	[75]
**24e**	−2.55	−5.11	4.40	10^7^	−0.5	[75]
**25**			0.014	10^3^		[76]
**26**			0.68	8 × 10^4^		[77]

**Table 3 nanomaterials-13-01750-t003:** Representative performance data of thiophene- and furan-containing macrocycles in p-type OFETs.

Material	LUMO (eV)	HOMO (eV)	*µ* (cm^2^ V^−1^ s^−1^)	*I*_on_/*I*_off_	*V*_T_/V	Ref.
**27a**	−3.29	−4.98	2.0 × 10^−2^	1.1 × 10^3^		[78]
**27b**			0.29	1.34 × 10^3^	−12.9	[80]
**27c**			0.63	3 × 10^2^	−7.47	[80]
**27d**		−5.04	0.23	5.27 × 10^5^	−17.7	[81]
**27e**		−4.98	9.6 × 10^−3^	2.42 × 10^5^	−56.1	[81]
**27f**		−5.07	0.73	1.4 × 10^7^	−4	[82]
**28a**			0.40	3.51 × 10^3^	20.0	[83]
**28b**			0.11	10^2^	24.4	[83]
**29**	−2.57	−4.83	1.92 × 10^−4^	10^2^	−1	[85]

**Table 6 nanomaterials-13-01750-t006:** Representative performance data of perylene imide-containing macrocycles in n-type OFETs.

Material	LUMO (eV)	HOMO (eV)	*µ* (cm^2^ V^−1^ s^−1^)	*I*_on_/*I*_off_	*V_T_*/V	Ref.
**44**	−3.87	−5.39	(1.5 ± 0.2) × 10^−3^			[117]
**45**	−3.90	−5.69	(1.5 ± 0.2) × 10^−3^			[117]
**46a**			6.8 × 10^−4^			[118]
**46b**			1.5 × 10^−2^			[118]
**47**			4.1 × 10^−3^			[119]
**48**			9.9 × 10^−4^			[119]

**Table 9 nanomaterials-13-01750-t009:** Representative performance data of cyclophane in OLEDs.

Material	HOMO/LUMO (eV)	*V*_on_/DV (V)	CE (cd A^−1^)	PE (lm W^−1^)	EQE (%)	Ref.
**73**	−5.56/−2.63	6.6/			1.52	[155]
**74**	−5.74/−2.68	6.4/	10.5	4	6.4	[156]
**75**	−5.71/−2.58	1.5/	2.5	0.71	1.5	[155]
**76**	−6.08/−2.72	3.97/	1.21	0.43	0.71	[157]
**77**			11.2		0.5	[158]
**78**		3.82/	9.93	8.25	2.82	[159]
**79**	−5.1/−1.75	6.7/			11	[160]
**80**		9.3/			0.2	[161]

**Table 10 nanomaterials-13-01750-t010:** Representative performance data of fluorene-containing macrocycles in OLEDs.

Material	HOMO/LUMO (eV)	*V*_on_/DV (V)	CE (cd A^−1^)	PE (lm W^−1^)	EQE (%)	Ref.
**81a**	−5.03/−2.26	13/	0.63			[162]
**81b**	−5.06/−2.28	14.1/	0.93			[162]
**82**	−5.07/−1.6	5.4/	22.6		5.3	[163]
**83**	−5.09/−0.95	6.0/	19.4	9.0	8.2	[164]
**84**	−5.27/−1.83	3.0/	17.7	11.9	7.7	[165]
**85**	−5.56/−2.19	3.3/	18.8	14.8	8.7	[166]
**86**	−5.35/−2.35	5.38/	0.83			[167]

**Table 11 nanomaterials-13-01750-t011:** Representative performance data of cyclo-meta-phenylenes in OLEDs.

Material	HOMO/LUMO (eV)	*V*_on_/DV (V)	CE (cd A^−1^)	PE (lm W^−1^)	EQE (%)	Ref.
**87a**		/8.7			13.9	[170]
**87b**	−6.2/−2.87	/5.7	94.1	43.5	22.8	[172]
**87c**	−5.95/−1.17	/5.1	88	54.4	24.8	[173]
**87d**	−6.0/−1.23	/5.8	67.3	36.0	18.7	[173]
**87e**	−6.9/−2.07	/8.3			9.9	[174]
**88a**		/8.6			13.2	[170]
**88b**	−6.04/−2.55	/4.5	30.1	17.3	7.3	[172]
**88c**	−6.0/−2.32	/5.6	7.9	29.2	7.9	[172]
**88d**	−5.91/−1.14	/5.5	50.1	28.6	14.2	[173]
**88e**	−5.96/−1.21	/4.8	21.7	11.4	5.3	[173]
**88f**	−6.88/−2.07	/8.0			6.3	[174]
**89**	−5.78/−2.29	/7.8		22.6	15.5	[171]

**Table 12 nanomaterials-13-01750-t012:** Representative performance data of arylamine- and triazine-containing macrocycles in OLEDs.

Material	HOMO/LUMO (eV)	*V*_on_/DV (V)	CE (cd A^−1^)	PE (lm W^−1^)	EQE (%)	Ref.
**90a**		2.0/			11.6	[175]
**90b**					10	[176]
**91**	−4.94/−2.31	3.6/	47.3	39.1	15.7	[177]
**92**		3.8/	53.7	37.0	17.1	[180]
**93**	−5.38/−2.66	3.5/	48.8	34.9	15.5	[180]

## Data Availability

Not applicable.

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
