# Peer review of "Structure–Property Relationship of Macrocycles in Organic Photoelectric Devices: A Comprehensive Review"

_nanomaterials, 2023, doi:10.3390/nano13111750_

Round 1
Reviewer 1 Report
The manuscript submitted by Ying Wei, Tonglin Yang, Linghai Xie and colleagues represents an interesting review on the use and application of macrocyclic compounds in organic photoelectric devices such as OFETs, OLEDs, OPVs and DSSCs.
Many examples of recent researches in the field combined with useful references to the older background are collected and discussed in the text in a way that I consider exhaustive and clear for the readers.
The selected topic is of interest and role and advantages of the macrocyclic structure in providing improved properties to the devices are underlined.
In my opinion the manuscript deserves publication in “Nanomaterials” after the following minor revisions:
Line 67: TiOPc is mentioned for the first time in the text, then it is necessary to report the extended name Titanyl phthalocyanine followed by the acronym (TiOPc)
Line 97: to refer to the different devices it is better to use the same annotation of the figure then replace i, ii, iii and iv with the letters a, b, c and d
Line 259: Compounds 5c and 5e are mentioned but there are no structures referred to them in any figure. Verify which compounds are
Line 281: systematic should be systematically
Line 306: Introducing the triarylamine derivatives, it is useful to recall the number of the figure wherein their structures are represented
Line 334: The acronym Cbz is typically used for the Carbobenzyloxy protecting group. It would be better use another one for carbazole to prevent a misunderstanding
Line 397: What does it mean 6? If it means the number of the studied devices, write it as word because as number can be misleading
Line 451: Figure 6 should be Figure 7
Line 530: Specify ITO in extended form. In general, check all the acronyms used because sometime they are reported without any explanation of their meaning. See also for PSS at line 542
Line 535-536: This sentence probably contains an error since it is not clear what it means
Line 644: what does CBP mean?
Line 705: and PVK?
Line 708: not clear, please modify that part of the sentence
Line 725: probably at least a word is missing and then the significance of the sentence is missing
Line 730-733: Also this sentence is not clear, please check it and rephrase
Figure 12: compounds 84 and 85 do not contain fluorene units, then they must be moved to another figure or the caption of figure 12 must be changed/completed accordingly
Line 959-961: this sentence is not necessary at this point of the text. Phthalocyanine have been widely described above and their definition already reported at line 112
Line 1000-1001: this sentence is not necessary at this point of the text. Perylene-based derivatives have been widely described above and the definition of perylene already reported at line 445
The quality of English is good enough and, apart few sentences reported in the comments that must be revised, the text can be rather fluently read
Reviewer 2 Report
This manuscript is a review paper, and it is important to state this explicitly in the title to accurately reflect the content.
The aim of this review is to provide a comprehensive analysis of various macrocycle structures, focusing on identifying the key factors that influence the structure-property relationship between macrocycles and their optoelectronic device properties. These factors include structural stability, rigid skeleton, inherent pore structure, absence of end-group effects, size-dependent optoelectronic properties, and fullerene-like charge transport characteristics. The review is interesting to the scientific community.
To present the analysis of the published work, a series of tables have been included. However, it would greatly benefit the reader if the data presented in the tables were complemented with figures that compare the relative performance of each material. These figures could showcase typical benchmark parameters, such as fill factor or field-effect mobility, plotted as a function of the material. This visual representation would aid the reader in assessing the materials' performance across different classes, such as increasing mobility.
The conclusions drawn in this paper are poor, as they lack clear guidelines for selecting the best materials. Expanding the conclusion section by providing specific and actionable guidelines is highly recommended.
In summary, before this work can be accepted for publication in Nanomaterials, I strongly recommend the incorporation of the following changes:
1- Modify the title of the manuscript to explicitly state that it is a review work.
2- Consider plotting some of the tabulated data in a graphical format, organizing devices/materials into classes on the X-axis, and representing relevant benchmark parameters on the Y-axis. This visual information will enhance the reader's ability to evaluate device and material performance.
3- Include comments on material stability, as it is often overlooked in this review.
4- Improve the conclusions section by providing guidelines for selecting the best materials for specific devices.
Additionally, there are some minor issues related to typos:
On line 451, there is a problem with the references: "[Error! Reference source not found.]"
On line 461, remove the italics from the value of mobility "45".
On line 686, there is a formatting problem in Table 8.
On line 878, there is an issue with the reference.
Reviewer 3 Report
The review is devoted to structure-property relationship of different types of macrocycles (phthalocyanines, porphyrins, cyclophane, etc.) investigation. The manuscript is interesting and contains new data about possible applications in organic photoelectronic devices, however it cannot be accepted in the present form and should be improved according to the following comments:
1. Some chemical terms are used inappropriately; the review contains many typos.
2. In my opinion, the introduction is not well written. There is no description of the problems of work, the advantages and disadvantages of a narrow range of macrocyclic compounds are described. A proper comparison with other analogues for the active layers of photoelectronic devices has not been carried out. It is not indicated for what period of time the review of articles was carried out (since 2000s or since 2010?).
3. There is no logic in the text. The text contains structurally complex sentences, which makes it difficult to understand the information. The review mainly consists of listing facts from a large number of articles, but there are practically no descriptions of any patterns and dependencies. In Conclusion, it is necessary to describe the structure-property relationship. How do the properties of the material depend on the molecular structure of the macrocycle, device configuration, sublayer, and other characteristics described in the review?
4. The review requires careful proofreading. In the text, spaces, dots were sometimes omitted, links were taken out of the sentence (after the dot). Line 125- empty brackets, something is missing, line 180 – wrong formula of 5b (OVPc4C8?), line 259 «The high mobilities of 5c and 5e»- what is 5c and 5e, line 417- 9.3×104 – wrong mobility value (power), etc.
5. Authors should carefully check the spelling of all units of measurement. Units of measurement must be written uniformly.
6. There are duplicate number of Ref. (For example, line 967, ref. from 181 to 186). There is no reference to the group of works by M. Bouvet and his co-workers devoted to phthalocyanine and porphyrins OFETs (doi.org/10.3390/s20174700, doi.org/10.1021/acssensors.0c00877, etc).
7. Some of the pictures are presented in poor quality, the signatures (number of compounds) are hard to see. ‘Bulk’ macrocycles should be redrawn, for example, in Figure 7 it is not clear what structure the compound has.
the review contains typos
Round 2
Reviewer 3 Report
I thank the authors for their responses to the reviwers comments. Now review is ready for publications in Nanomaterials.